# Population-based sequencing of *Mycobacterium tuberculosis* reveals how current population dynamics are shaped by past epidemics

Irving Cancino-Muñoz[1†], Mariana G López[1*†], Manuela Torres-Puente[1], Luis M Villamayor[2], Rafael Borrás[3], María Borrás-Máñez[4], Montserrat Bosque[5], Juan J Camarena[6], Caroline Colijn[7], Ester Colomer-Roig[2,6], Javier Colomina[3], Isabel Escribano[8], Oscar Esparcia-Rodríguez[9], Francisco García-García[10], Ana Gil-Brusola[11], Concepción Gimeno[12], Adelina Gimeno-Gascón[13], Bárbara Gomila-Sard[14], Damiana Gónzales-Granda[15], Nieves Gonzalo-Jiménez[16], María Remedios Guna-Serrano[12], José Luis López-Hontangas[11], Coral Martín-González[17], Rosario Moreno-Muñoz[14], David Navarro[3], María Navarro[18], Nieves Orta[17], Elvira Pérez[19], Josep Prat[20], Juan Carlos Rodríguez[13], Ma Montserrat Ruiz-García[16], Hermelinda Vanaclocha[19], Valencia Region Tuberculosis Working Group, Iñaki Comas[1,21*]

[1]Tuberculosis Genomics Unit, Instituto de Biomedicina de Valencia (IBV-CSIC), Valencia, Spain; [2]Unidad Mixta "Infección y Salud Pública" (FISABIO-CSISP), Valencia, Spain; [3]Microbiology Service, Hospital Clínico Universitario, Valencia, Spain; [4]Microbiology and Parasitology Service, Hospital Universitario de La Ribera, Alzira, Spain; [5]Microbiology Service, Hospital Arnau de Vilanova, Valencia, Spain; [6]Microbiology Service, Hospital Universitario Dr Peset, Valencia, Spain; [7]Department of Mathematics, Faculty of Science, Simon Fraser University, Burnaby, Canada; [8]Microbiology Laboratory, Hospital Virgen de los Lirios, Alcoy, Spain; [9]Microbiology Service, Hospital de Denia, Denia, Spain; [10]Computational Genomics Department, Centro de Investigación Príncipe Felipe, Valencia, Spain; [11]Microbiology Service, Hospital Universitari i Politècnic La Fe, Valencia, Spain; [12]Microbiology Service, Hospital General Universitario de Valencia, Valencia, Spain; [13]Microbiology Service, Hospital General Universitario de Alicante, Alicante, Spain; [14]Microbiology Service, Hospital General Universitario de Castellón, Castellón, Spain; [15]Microbiology Service, Hospital Lluís Alcanyis, Xativa, Spain; [16]Microbiology Service, Hospital General Universitario de Elche, Elche, Spain; [17]Microbiology Service, Hospital Universitario de San Juan de Alicante, Alicantes, Spain; [18]Microbiology Service, Hospital de la Vega Baixa, Orihuela, Spain; [19]Subdirección General de Epidemiología y Vigilancia de la Salud y Sanidad Ambiental de Valencia (DGSP), Valencia, Spain; [20]Microbiology Service, Hospital de Sagunto, Sagunto, Spain; [21]CIBER of Epidemiology and Public Health (CIBERESP), Madrid, Spain

*For correspondence:
mglopez@ibv.csic.es (MGL);
icomas@ibv.csic.es (IC)

†These authors contributed equally to this work

Group author details:
Valencia Region Tuberculosis Working Group See page 13

**Abstract** Transmission is a driver of tuberculosis (TB) epidemics in high-burden regions, with assumed negligible impact in low-burden areas. However, we still lack a full characterization of transmission dynamics in settings with similar and different burdens. Genomic epidemiology can greatly help to quantify transmission, but the lack of whole genome sequencing population-based

studies has hampered its application. Here, we generate a population-based dataset from Valencia region and compare it with available datasets from different TB-burden settings to reveal transmission dynamics heterogeneity and its public health implications. We sequenced the whole genome of 785 *Mycobacterium tuberculosis* strains and linked genomes to patient epidemiological data. We use a pairwise distance clustering approach and phylodynamic methods to characterize transmission events over the last 150 years, in different TB-burden regions. Our results underscore significant differences in transmission between low-burden TB settings, i.e., clustering in Valencia region is higher (47.4%) than in Oxfordshire (27%), and similar to a high-burden area as Malawi (49.8%). By modeling times of the transmission links, we observed that settings with high transmission rate are associated with decades of uninterrupted transmission, irrespective of burden. Together, our results reveal that burden and transmission are not necessarily linked due to the role of past epidemics in the ongoing TB incidence, and highlight the need for in-depth characterization of transmission dynamics and specifically tailored TB control strategies.

## Editor's evaluation

This work presents insightful epidemiologic and phylogenetic analyses of tuberculosis cases across Valencia, Spain, and comparator low-burden (Oxfordshire, UK) and high-burden (Karonga, Malawi) regions. Findings reveal that the "low burden" observed in Valencia is not in fact reflective of low transmission in this setting, with detected lineages likely to have circulated locally over the course of decades and to have been transmitted in the community.

## Introduction

Tuberculosis (TB) has been the first cause of death by an infectious disease for years surpassing HIV according to the World Health Organization (WHO). In 2019 were reported 10 million new TB cases and 1.4 million deaths, with these numbers likely to increase due to the COVID-19 pandemic (*Glaziou, 2020*). Recognizing heterogeneity across settings in the population-level dynamics of TB is key to advance to new stages in local and global TB control (*Mathema et al., 2017*). Recent transmission significantly contributes to the global TB-burden mostly in the high incidence regions and its control is imperative to achieve the goal of the End TB Strategy (*Auld et al., 2018*; *Guerra-Assunção et al., 2015*; *Yates et al., 2016*).

On the contrary, in low-burden countries the assumption is that transmission plays a minor role, supported by the fact that in countries close to the pre-elimination phase (<5/100,000 cases) TB cases are mainly contributed by re-activations of latent TB infection (LTBI) from recently arrived migrants (*Menzies et al., 2018*). However, whether the minor role of transmission in pre-elimination phase countries can be extrapolated to other low-burden countries is currently unknown. Understanding the correlation between burden and transmission and country specific dynamics is key if tailor-made control strategies are needed.

Measuring transmission is still challenging. Genomic epidemiology, based on comparative pathogen genomics, has been successfully applied in some settings, but usually not at a population scale, needed to profile the transmission dynamics in a setting. Using genomic epidemiology is not exempt from limitation, e.g., as transmission cases associated with LTBI are missed as well as cases without culture (see *López et al., 2020*). However, it allows us to compare transmission clustering rates and dynamics across settings in a standard way. A common approach to characterize transmission with whole genome sequencing (WGS) is to use pairwise single nucleotide polymorphisms (SNPs) distances (*Gardy et al., 2011*; *Tagliani et al., 2021*; *Walker et al., 2013*). The WGS displays higher resolution, provides accurate results tracking recent transmission (*Jajou et al., 2018*; *Marais et al., 2017*; *Meehan et al., 2019*; *Nikolayevskyy et al., 2019*), and reports greater agreement with epidemiological results than previous molecular markers (*Diel et al., 2019*; *Meumann et al., 2021*; *Nikolayevskyy et al., 2016*; *Roetzer et al., 2013*). In addition, Bayesian dating allows us to correlate genetic differences between strains and time of transmission clusters (*Meehan et al., 2018*). Even more, the higher resolution of genomic data also allows us to go beyond standard clustering of cases and can reveal individual transmission links (TLs) and timing of events (*Xu et al., 2019*).

Despite WGS reliability, there exists controversy regarding the SNP threshold employed to delineate genomic clusters (gClusters). A cut-off of 5 SNPs has been widely accepted for the clustering of recently linked cases (*Meehan et al., 2019*; *Nikolayevskyy et al., 2019*) while an upper value of 12 SNPs also incorporates older transmission events (*Walker et al., 2013*); however, the extent to which the identification of those cases can aid epidemiological investigations remains controversial (*Bjorn-Mortensen et al., 2016*; *Jajou et al., 2018*). It is also unclear the extent to which those cutoffs apply to all settings given differences in social, host, and pathogen factors across settings. Even if universal, understanding transmission dynamics goes beyond recent transmission events, which have an actionable value for public health, but that do not capture the long-term dynamics in a population.

The lack of WGS studies at the population level represents the main limitation to the validation of these thresholds across clinical settings, and to understand the transmission dynamics in different areas. Here we use available datasets from a low burden setting (Oxfordshire, incidence 8.4 cases per 100,000 inhabitants) and from a high burden setting (Malawi, incidence 87 cases per 100,000 inhabitants) and compare to a newly generated dataset in Valencia region (incidence ~8 cases per 100,000 inhabitants).

In the Valencia region, the contribution of recent transmission to local TB burden remains largely unknown. First, we investigated the epidemiology and dynamics of TB transmission in the Valencia region, the fourth most populated region of the country, over 3 years. Second, we evaluated the general use of an SNP threshold in cluster definition and public health investigations, in this particular setting. Third, we compared the transmission dynamics in the Valencia region with similar population-based studies from locations with different TB burdens (*Guerra-Assunção et al., 2015*; *Walker et al., 2014*). Our results demonstrate that current TB incidence in Valencia region mainly derives from sustained transmission similar to a high-burden setting. Comparison among settings highlight little correlation between burden and transmission and suggest the need of tailor-made control strategies.

## Materials and methods
Extended and detailed methods in Appendix 1.

### Sample selection and study design
About 1388 TB cases were reported between 2014 and 2016 by the Valencian Regional Public Health Agency (DGSP), 1019 with positive culture. All the available (785) samples were collected from 18 regional hospitals (*Appendix 1—figure 1*). Demographic, clinical, and microbiological records were obtained from the routine TB surveillance system, for 724 of the total samples. All diagnosed TB-positive patients completed a standardized questionnaire provided by the DGSP. In addition, conventional contact tracing is conducted for most cases as per WHO guidelines (https://www.sp.san.gva.es/Dgsp-Portal/docs/GuiaTuberculosis2008.pdf).

*M. tuberculosis* structure and clustering analysis were performed with the total sequences. Epidemiological and transmission dynamics analysis were carried on with the samples with available information (724).

Approval for the study was given by the Ethics Committee for Clinical Research from the Valencia Regional Public Health Agency (*Comité Ético de Investigación Clínica de la Dirección General de Salud Pública y Centro Superior de Investigación en Salud Pública*). Informed consent was waived on the basis that TB detection forms part of the regional compulsory surveillance program of communicable diseases. All personal information was anonymized, and no data allowing patient identification was retained.

### DNA extraction and sequencing
Clinical isolates were cultured in Middlebrook 7H11 agar plates supplemented with 10% OADC (Becton-Dickinson) for 3 weeks at 37°C. After an inactivation step (90°C, 15 min), DNA was extracted using the cetyl trimethyl ammonium bromide method from a representative sample from each patient (four-time plate scraping). All procedures were conducted in a biological safety level 3 laboratory under WHO protocol recommendations. Sequencing libraries were constructed with a Nextera XT DNA library preparation kit (Illumina, San Diego, CA), following the manufacturer's instructions. Sequencing was performed using the Illumina MiSeq platform.

## Bioinformatics analysis

Data analysis was carried out following a validated previously described pipeline (http://tgu.ibv.csic.es/?page_id=1794, *Meehan et al., 2019*). Sequencing reads were trimmed with fastp (*Chen et al., 2018*), and kraken software (*Wood and Salzberg, 2014*) was then used to remove non-*M. tuberculosis* complex (MTBC) reads. Filtered reads were mapped to an inferred MTBC common ancestor genome (https://doi.org/10.5281/zenodo.3497110) using BWA (Burrows-Wheeler Aligner, *Li and Durbin, 2009*). SNPs were called with SAMtools (*Li, 2011*) and VarScan2 (*Koboldt et al., 2012*). GATK HaplotypeCaller (*McKenna et al., 2010*) was used for calling InDels. SNPs with a minimum of 10 reads (20×) in both strands and minimum base quality of 25 were selected and classified based on their frequency in the sample as fixed (>90%) or low frequency (10–89%). InDels with less than 20× were discarded. SnpEff was used for SNP annotation using the H37Rv annotation reference (AL123456.2). Finally, SNPs falling in genes annotated as PE/PPE/PGRS, 'maturase,' 'phage,' and '13E12 repeat family protein'; those located in insertion sequences; those within InDels or in higher density regions (>3 SNPs in 10 bp) were removed due to the uncertainty of mapping. Next, variants were compared with recently published catalogs with validated association between mutations and phenotypic resistance (*Ngo and Teo, 2019*) in order to predict high-confidence resistance profiles to first- and second-line drugs. Lineages were determined by comparing called SNPs with specific phylogenetic positions established (*Coll et al., 2014*; *Stucki et al., 2016*). An in-house R script was used to detect mixed infections based on the frequency of lineage- and sublineage-specific positions (*López et al., 2020*). Read files were deposited in the European Nucleotide Archive (ENA) under the bioproject numbers PRJEB29604 and PRJEB38719 (*Supplementary file 1*). Sequences from two population-based studies in Oxfordshire (*Walker et al., 2014*), with 92% of culture-positive cases sequenced, and Malawi (*Guerra-Assunção et al., 2015*), with 72% of culture-positive cases sequenced, were downloaded from ENA and analyzed as for the sequences generated in this study. All the custom scripts used are available in https://gitlab.com/tbgenomicsunit/ThePipeline. (Copy archieved at swh:1:rev:a725827cb664e6d995823f3f30fcd-1d7e16f63d2, *Belda-Álvarez, 2022*).

## gClustering and phylogenetic analyses

The pairwise SNP distance was computed with the R *ape* package. The gCluster were constructed if the genetic distance between at least two patients' isolates fell below a defined threshold. Cluster monophyly was confirmed in a maximum likelihood tree (50,184 SNPs).

Timed phylogenies were inferred with Beast v2.5.1 (*Bouckaert et al., 2014*). Ancient TB DNA (*Bos et al., 2014*) and samples from a recent Spanish outbreak were included as calibration data. We constructed SNP alignments for each dataset, excluding known variants related to drug resistance, then we corrected the ascertainment bias by adjusting the clock rate to incorporate invariants sites (*Supplementary file 2*). Extended methodology and comparison with other ascertainment bias methods is detailed in Appendix 1. Dating was performed using GTR + GAMMA substitution model (General Time Reversible with gamma distributed rates heterogeneity) , a strict molecular clock model, and a coalescent constant size demographic model, as previously described (*López et al., 2020*). Three independent runs of Markov Chain Monte-Carlo length chains of 10 million were performed. Adequate mixing, convergence, and sufficient sampling were assessed in Tracer v1.6, after a 10% burn-in.

## Tracking historical TLs

Historical TLs were defined as nodes or tree bifurcations occurring over time phylogenies (*Appendix 1—figure 2*). The rationale for this approach is based on the assumption that if few pathogen mutations are expected to be observed during a host's infection, as is the case of *M. tuberculosis*, lineages split only at transmission (*Hall et al., 2016*). In addition, as *M. tuberculosis* is an obligate pathogen, every strain is by definition, linked to others by recent or historical transmission events. In this sense, each node in the phylogeny represents a minimum one, and likely many, person-to-person transmission. Tips in the tree are the result of decades of transmission or, which is the same, multiple transmission events occurring along the branch; however, most of the secondary cases generated are missing, as they do not develop active TB or were cured before sampling (among other reasons, *Appendix 1—figure 2A*). Thus, only those lineages persisting until sampling time were recovered (*Appendix 1—figure 2B*). Moreover, the greater the transmission and the more sustained over time, the greater the

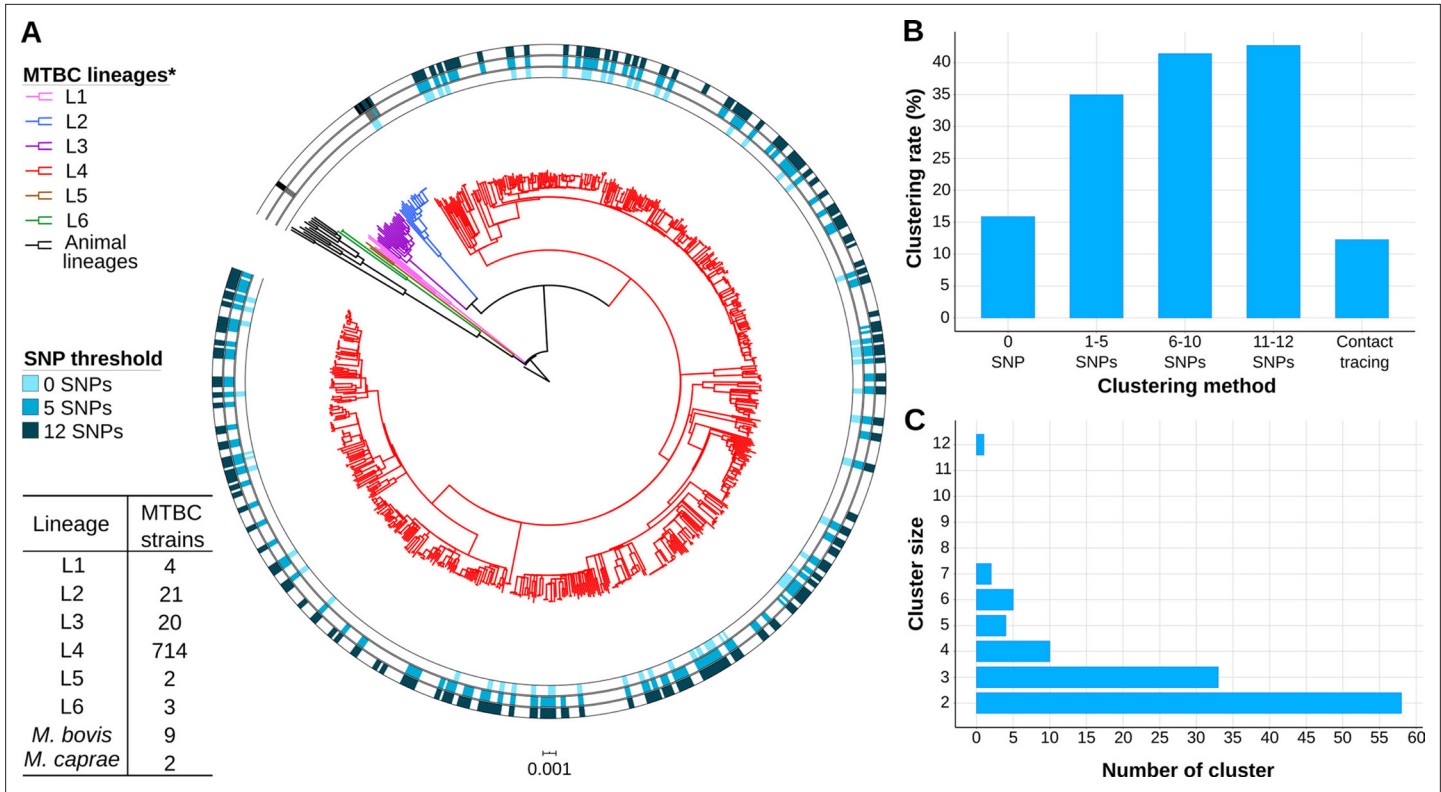

**Figure 1.** Genomic characterization of the study region. (**A**) Phylogeny of 775 tuberculosis (TB) isolates collected during the years 2014 and 2016. Each ring represents genomic clusters detected by different single nucleotide polymorphism (SNP) thresholds (0, 5, and 12 SNPs). *Mycobacterium canneti* was used as an outgroup. (**B**) Clustering percentage, i.e. percentage of samples within clusters for different SNP thresholds. (**C**) Number of genomic clusters by different cluster sizes. A 12 SNP threshold was used as a standard. Cluster sizes of 8–11 samples were not detected. *Nomenclature proposed by **Comas et al., 2013**.

The online version of this article includes the following source data for figure 1:

**Source data 1.** Genomic cluster types; Spanish: clusters including only Spanish-born cases; foreign: clusters including only foreign-born cases; mix: clusters including Spanish and foreign-born cases.

chance of recovering secondary cases today. In addition, to estimate the time of the TLs, we used the time of the node, or the common ancestor, as this is the lower bound when the strain starts to circulate. Thus, each TL summarizes the number and time of the transmission along each tree branch. This analysis was conducted in the time trees generated with Beast, including both local and foreign cases; in order to avoid introductions we excluded nodes in which foreign samples appeared as ancestors, and only counted nodes including local-born tips occurring within 150 years before 2016 (yB 2016).

This analysis does not intend to define the direction of transmission or the exact moment when it occurred, as can be done with TransPhylo (**Didelot et al., 2017**), but instead to profile ancestral TLs through time and trace since when, the lineages recovered nowadays, have been circulating. Note that the concept 'TL' in this context does not indicate person-to-person contagion, instead it is the summary of multiple contagions occurring in a period of time, and indicates that a particular extant strain has been involved in transmission during that period.

## Results

### *M. tuberculosis* population structure and demographic characteristics in Valencia region

We sequenced 77% of the TB culture-positive cases reported between 2014 and 2016 in Valencia region with a 4.3 million population. Around 10 samples were removed as non-MTBC isolates or likely

mixed infections (*Supplementary file 1*). We identified 6 different lineages (L) circulating in the region (*Coll et al., 2014*; *Stucki et al., 2016*), with L4 the most frequent (92.1%) (*Figure 1A*).

Characteristics of TB cases are summarized in *Supplementary file 3*, reporting the sequenced samples as a representative subset of the total culture-positive cases. TB incidence in the region ranged between 8.3 and 8.7 with higher incidence in migrants (mean 23.6) than in local born individuals (mean 6.9). Detailed epidemiological analysis of TB cases in Valencia region is presented in *Supplementary file 3* and *Supplementary file 4*, remarkably 63% of all cases were Spanish-born patients, while 30% came from high-incidence countries, and 7% from other low-incidence countries. When we observed risk factors associated with TB, we found that diabetes was present in 10.4% of cases; although this was more prevalent in Spanish-born patients (OR 2.7, CI 1.5–5.4, p<0.001), values were similar to diabetes prevalence in the general Spanish population.

## Epidemiological and gClustering

Conventional contact tracing investigations (see Methods) identified 66 epidemiological clusters, including 97 cases, accounting for 12.5% of transmission in the Valencia region (*Figure 1B*). Considering the widely used pairwise distance threshold of 12 SNPs for defining transmission, we identified 112 gClusters, including 331 (42.7%) patients, with cluster size ranging from 2 to 12 cases (*Figure 1C*, *Figure 1—source data 1*). Although these clusters included foreign-born patients, Spanish-born patients were more likely part of genomically linked groups (OR 2, CI 1.44–2.79, p<0.001). In this regard, 42 gClusters exclusively comprised Spanish-born patients and 8 included only foreign-born patients.

In addition to Spanish origin, pulmonary localization of TB (OR 2.5, CI 1.60–3.98, p<0.001), and younger age also appeared associated with clustering by Fisher's exact test. After correcting for confounders using a logistic regression, Spanish origin remains significantly associated with clustering (p<0.001); younger age, pulmonary localization of TB, and male sex were also significant (p<0.05, *Supplementary file 4*). Finally, 90% of TB cases in Valencia region are susceptible to all antibiotics used in treatment, so resistance has no impact on clustering.

We also assessed gClusters considering lower SNP thresholds, and observed that independently of the cut-off considered, the clustering rate obtained by contact tracing was always lower than the

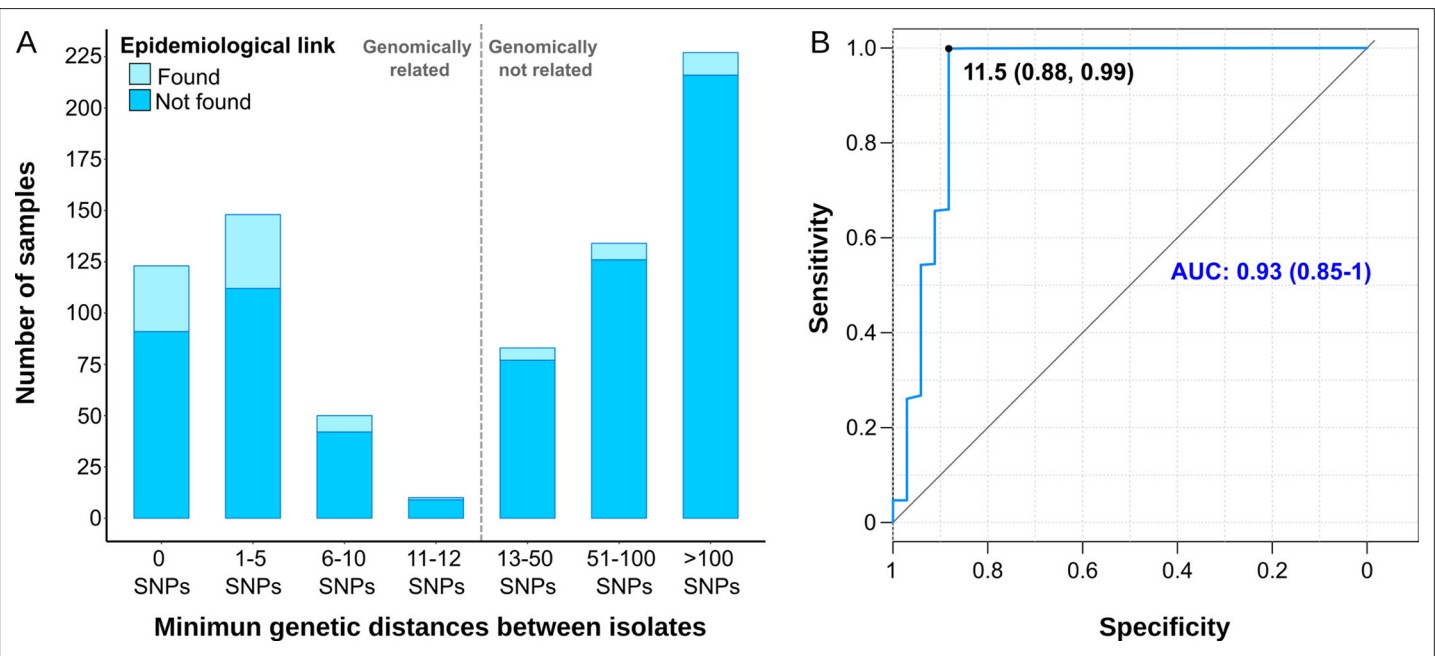

**Figure 2.** Comparison between epidemiological and genomic clustering. (**A**) Clustered samples using different pairwise distance thresholds, bars denote the number of cases within clusters for each single nucleotide polymorphism (SNP) threshold. Gray dashed line separates the genomically linked samples (clustered) from those unlinked. (**B**) ROC (Receiver Operating Characteristics) curve for different pairwise distance thresholds between 0 and 2000 SNPs, indicating the optimal SNP cut-off values with its correspondent specificity and sensitivity values, the area under the curve (AUC), and its confidence intervals.

genomic estimates (*Figure 1B*). A high number of genomic links were not detected by epidemiological investigation, while some epidemiological links were not corroborated by any gClustering threshold (*Figure 2A*). Comparison of both approaches revealed that only 15.4% of the 331 patients within gClusters (12 SNPs) had an identified epidemiological link (Appendix 1, *Supplementary file 5*).

We benchmarked WGS as a tool to quantify transmission against contact tracing, using the latter as the gold standard (*Diel et al., 2019*). In general, as the SNP threshold decreases, sensitivity diminishes, but specificity and accuracy increases (*Supplementary file 6*). A ROC curve establishes 11.5 SNPs as the optimal value for the SNP cut-off that maximizes the agreement between epidemiological investigation and genomic data, with an area under the curve higher than 0.9 (*Figure 2B*). Then, we used 12 SNPs threshold to define clusters in the following analyses.

## Genetic thresholds for transmission are not universal across settings

Next, we calculated the percentage of Spanish-born cases clustered by a range of pairwise distances (0–150 SNPs) and compared with the clustering of local cases in other settings (*Guerra-Assunção et al., 2015*; *Walker et al., 2014*), where more than 70% of all culture-positive cases were sequenced. We observed a bimodal pattern for Oxfordshire, with the transmission groups clearly differentiated from the other unlinked cases with distances higher than 50 SNPs. These findings agree with the 12 SNP value proposed as a means to identify recent transmission in low-burden settings (*Walker et al., 2014*). For the Valencia region and Malawi, strains group in a large range of distance thresholds (SNPs 0–150). Thus, there exists a continuous clustering throughout the distance values. The results strongly suggest that a strict transmission threshold of 12 SNPs (or any other threshold) does not apply to all the settings when analyzing transmission dynamics, particularly in those places with higher transmission burdens (*Figure 3A*), and specially if we want to understand long-term transmission (i.e. the survival and expansion of particular clones/strains in a population). However, strict SNP thresholds are informative to health authorities (see Discussion).

## Age of local gClusters at different SNP thresholds and impact on public health

Next, we evaluated how old are the gClusters identified by the standard 12 SNP threshold. Thus, we inferred the age of the local gClusters for the three settings. Dating results of the youngest and the oldest gClusters are summarized in *Table 1*, while complete results are detailed in *Figure 3—source data 1–3*. We can trace gClusters 31 years back from the most recent sample collected for both the Valencia region and Malawi; however, we only retrieved samples that formed part of gClusters, 19 years before the most recent Oxfordshire sample. The alternative time calibration samples included (Appendix 1) displayed similar results, thereby allowing comparisons among datasets. Our inference of clusters' ancestors are in agreement with previous studies, using a similar Bayesian approach, and defining a timespan of up to 10 years for 5 SNP cut-offs (*Meehan et al., 2018*). Thus many genomic links based on 12 SNP distance are beyond the action of public health interventions. Using data from Valencia region, we further investigated the role of genomic distances in public health by evaluating the age of epidemiologically linked cases. Most of the epidemiologically linked cases have a common ancestor less than 10 years before the most recent sample, and the distance between samples typically ranged between 0 and 4 SNPs, with only one individual link separated by 11 SNPs (*Supplementary file 5*). While the ROC curve indicated a 12 SNP threshold to capture most epidemiological links, the reality is that strains linked by more than 5 SNP are beyond the action of public health interventions as they involve too old TLs. Our results imply that events useful for public health investigations are better captured by a 5 SNP threshold even though some epidemiological links are missing. But the reverse is also true, and more dramatic. Even when using a 5 SNP threshold public health only identifies around 15% of the cases in gClusters. This holds true even for pairs of isolates with 0 SNP differences. The many genomic links missed by public health investigations in Valencia region reminds of what is seen in high-burden countries, where contagion occurs outside the traditional household or work settings.

## Historical TLs between clinical settings highlight distinct epidemic dynamics

In order to evaluate transmission dynamics in a setting over time, we defined historical TLs as the common ancestor of two circulating strains up to 150 yB 2016. To notice, we did not try to quantify

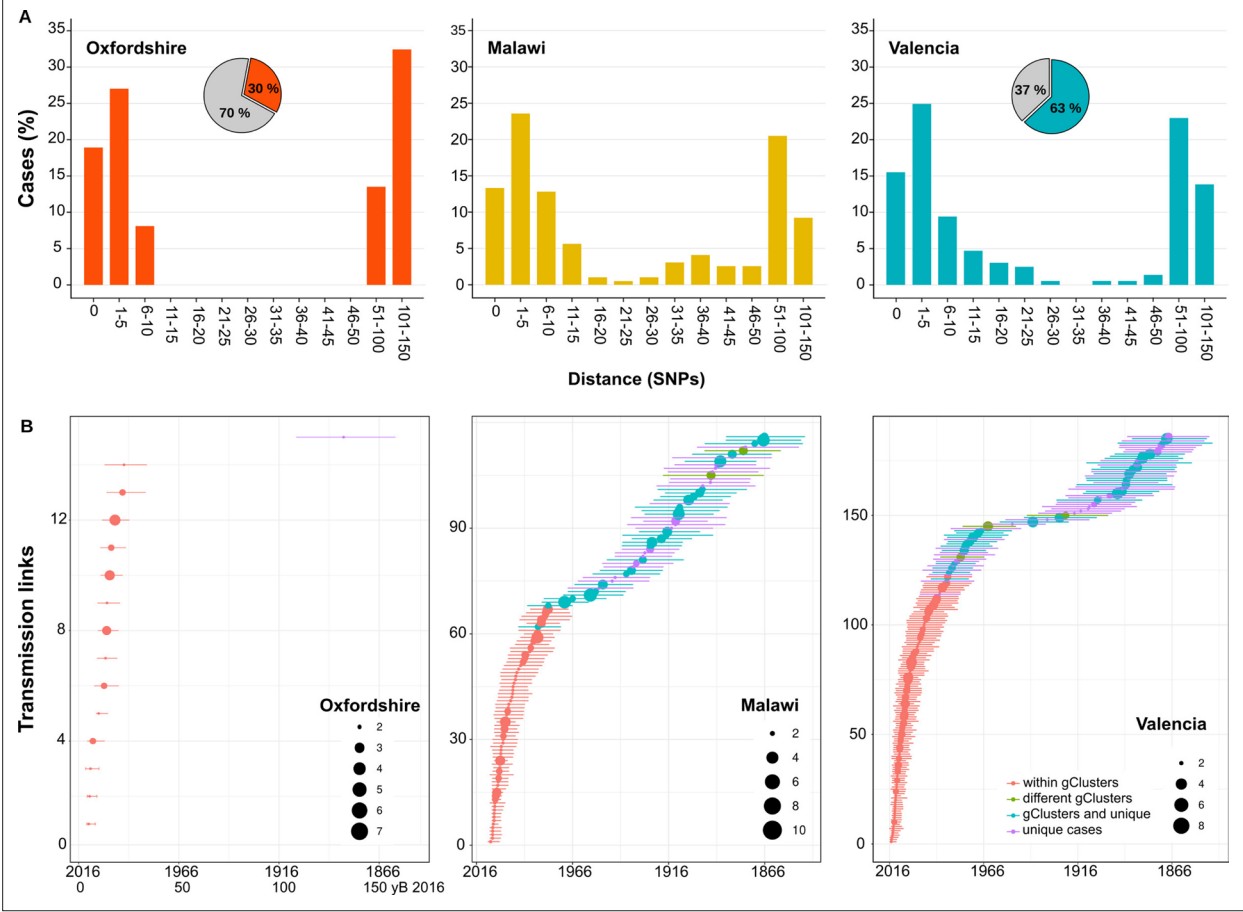

**Figure 3.** Historical transmission dynamics analysis. (**A**) Distribution of local-born cases clustered by different pairwise distance SNP thresholds. Cases are expressed as the percentage of the plotted samples. Pie charts represent the proportion of local-born (color) and foreign-born (gray) cases in each dataset. (**B**) Age of local transmission links over time in each setting. Circles represent median time, and lines represent 95% high probability density for each transmission link counted. Circle size represents the number of samples included in the corresponding link. Red denotes those transmission links including only samples within the same genomic transmission clusters (gClusters), green denotes links involving samples from different gClusters, blue denotes samples within gClusters and unique, and purple denotes unique cases. All links were obtained from *Figure 3—figure supplements 1–6* and are summarized in *Figure 3—source data 1–6*.

The online version of this article includes the following source data and figure supplement(s) for figure 3:

**Source data 1.** Bayesian dating results for all clusters (CL) from Oxfordshire.

**Source data 2.** Number of transmission links (TLs) traced back to 150 years before 2016 for Oxfordshire dataset.

**Source data 3.** Bayesian dating results for all clusters (CL) from Valencia region.

**Source data 4.** Number of transmission links (TLs) traced back to 150 years before 2016 for Oxfordshire dataset.

**Source data 5.** Number of transmission links (TLs) traced back to 150 years before 2016 for Malawi dataset.

**Source data 6.** Number of transmission links (TLs) traced up to 150 years before 2016 for Valencia dataset.

**Figure supplement 1.** Bayesian time tree for Oxfordshire dataset.

**Figure supplement 2.** Bayesian time tree for Malawi dataset.

**Figure supplement 3.** Bayesian time tree for Valencia region dataset.

**Figure supplement 4.** Bayesian time tree for Velncia Region dataset.

**Figure supplement 5.** Bayesian time tree for Valencia Region dataset.

**Figure supplement 6.** Bayesian time tree for Valencia Region dataset.

**Table 1.** Dating of local genomic clusters (gCluster).

Times of the oldest and youngest local gClusters obtained by a Bayesian analysis are presented, with values in years (AD) and 95% highest posterior density given in brackets. The number of gClusters and clustering percentage is provided for each dataset. The median distance ranges for all gClusters are also detailed.

| Dataset | Sampling period | Local samples | N local gCluster | Local clustering | Median distance range | Oldest gCluster | Youngest gCluster |
|---|---|---|---|---|---|---|---|
| Oxfordshire | 2006–2012 | 74 | 6 | 27% | 0–7 | 1993 (1982–2003) | 2009 (2003–2012) |
| Malawi | 2008–2010 | 106 | 40 | 49.80% | 0–14 | 1979 (1968–1988) | 2009 (2004–2010) |
| Valencia region | 2014–2016 | 456 | 65 | 47.40% | 0–11 | 1985 (1972–1996) | 2015 (2012–2016) |

how many transmission events have happened over the last 150 years. Our rationale is that many person-to-person transmission events likely occurred along branches between nodes or nodes and terminals, they are impossible to quantify, but we can summarize all these events as one TL, as we are confident that at least one transmission event occurred along the branch. The exact time of the transmission is not possible to estimate either, instead our rationale is that when two circulating strains had a common TL in the past, this ancestor represents a lower-bound for when the strains started to circulate. Thus, we compare how many links have occurred during a period of time among different settings, as an approach of long term transmission dynamics analysis. In our approach, we only considered genomic data from local-born patients to avoid the influence of imported genotypes in our analysis.

Then, we counted the TLs in different time windows, in the three settings evaluated. In the case of Oxfordshire, we identified 14 links between 5 and 25 yB 2016, with the next TL being inferred between 100 and 150 yB 2016 (*Figure 3B*, *Figure 3—figure supplement 1*, *Figure 3—source data 4*). Thus, a gap of 75 years occurs between the most recent and the oldest TLs, explaining why the 12 SNP threshold performs well in this setting as a transmission marker. In the case of Malawi, we counted 70 links dating back to 50 yB 2016 and 46 dated between 50 and 150 yB 2016 (*Figure 3B*, *Figure 3—figure supplement 2*, *Figure 3—source data 5*). For the Valencia region, we counted 143 links that dated back 50 yB 2016 and 43 between 50 and 150 yB 2016 (*Figure 3B*, *Figure 3—figure supplements 3–6*, *Figure 3—source data 6*). The gap detected in Oxfordshire is not observed in Malawi or Valencia.

In the above analysis we had two strong assumptions. First, that the historical link shared by two strains happened in the setting under study and not elsewhere. We do believe this is the case as we only considered links involving local-born cases, thus minimizing the impact of importation/exportation in the analysis. In addition, a local origin is the most likely geographic location of historical TLs when analyzed in the context of a representative sample of global diversity (*Figure 3—source data 6*). Second, that two strains not only shared a historical link but also their sampling today reflect continuous transmission, not reactivation from a remote infection. Recent reanalysis of annual risk of infection in TB settings (*Dowdy and Behr, 2022*) as well as the incubation period (*Behr et al., 2018*) suggests that most cases of TB are due to recent transmission. Likewise, we reasoned that if old reactivations contribute to strains in the Valencia region, we should see an increment in the age of the TB patients belonging to the older clusters (i.e. patients infected 20 years ago and have reactivated recently). We found no difference when comparing the age of the patients belonging to a gCluster with the inferred age of the cluster (Welch two-samples t-test, p-values>0.1, *Appendix 1—figure 3*, *Supplementary file 7*), suggesting that the strains included in this study do not represent reactivations and that uninterrupted transmission is the most likely explanation for the old links observed.

## Discussion

Here, we present the first population-based study of TB transmission in Spain based on WGS. We sequenced the whole genome of a representative proportion of all the TB notified cases in the Valencia region that provides an accurate picture of the bacterial population structure, during 3 years. We

exhaustively researched TB transmission linked to local epidemiological data and, by comparing to other settings, highlighted four main characteristics defining dynamics and influence on TB incidence.

## Transmission can play a significant role in low-burden countries, especially among local-born patients

The percentage of genomically linked cases (12 SNPs) of around 43% in the total population, increases to 47% among the Spanish-borns being 31% among imported cases, suggesting that transmission among locally born patients majorly contributes to disease burden. These percentages remain high when considering a stricter threshold of 5 SNPs for clustering (35 vs. 39%, respectively). We found higher transmission in the Valencia region compared to other population based studies conducted in low-burden settings, where clustering rates ranged between 14 and 16% (*Jajou et al., 2018*; *Walker et al., 2014*) and somewhat closer to that reported in mid and high TB-incidence settings (39–85%) (*Guerra-Assunção et al., 2015*; *Gygli et al., 2021*; *Khan et al., 2019*; *Saavedra et al., 2022*; *Walker et al., 2022*; *Yang et al., 2022*; *Yates et al., 2016*). High transmission among Spanish-born individuals is a major contributor to disease burden in Valencia. By contrast, reactivation of infections in imported cases from high-burden settings is the significant driver in other low-burden settings (*Jajou et al., 2018*; *Kamper-Jørgensen et al., 2012*; *Meumann et al., 2021*; *Walker et al., 2014*). Thus, our results reveal the heterogeneity of the TB epidemic among settings, highlighting the lack of correlation between a region's TB burden and the level of local transmission.

## Community transmission can majorly contribute to TB cases in a low burden setting

In low-burden TB settings, comparison between contact tracing and WGS revealed that between 38 and 57% of genomically linked cases had also an epidemiological link (*Diel et al., 2019*; *Jajou et al., 2018*; *Walker et al., 2014*). In high-burden settings, which suffer from rampant community transmission (*Yates et al., 2016*), the agreement between both approaches is significantly lower (8–19%) (*Auld et al., 2018*; *Middelkoop et al., 2015*; *Verver et al., 2004*). In the Valencia region, we observed an agreement similar to high-burden settings (15.4%), meaning that almost 80% of transmission is missing by the health system, despite contact tracing being conducted in 78% of cases. As has suggested for high-burden settings, contact tracing among household close contacts will not have a significant effect on TB incidence at a community level (*McCreesh and White, 2018*; *Surie et al., 2017*), since much of transmission may result from casual contact in community settings between individuals not known to one another (*Auld et al., 2018*; *Guerra-Assunção et al., 2015*) and also, transmission associates more with social drivers, including the ways in which individuals interact and congregate (*Andrews et al., 2014*; *Mathema et al., 2017*). Thus, our results support that community transmission is behind the lack of agreement between genomic and epidemiological clusters observed in the Valencia region, and highlights its relevance in low burden settings.

## Genomic links are older than epidemiological links

The Valencia region's oldest gClusters dated to around 30 years before the sampling period. When considering only strains epidemiologically linked, the oldest most recent common ancestor can be traced less than 10 years. Thus, a 12 SNP threshold identifies both recent and older transmission events. A 5 SNP threshold dates clusters between 1999 and 2015 in agreement with recent transmission rendering more actionable results for public health, as was previously shown (*Jajou et al., 2018*; *Meehan et al., 2018*). However, a 5 SNP threshold still misses a percentage of cases linked by epidemiological data and vice versa, highlighting transmission complexity and the relevance of understanding its dynamics in each setting. Thus, a strict threshold has limitations and communicating a range, incorporating degrees of confidence, will be more valuable for public health interventions. This is particularly true in settings where transmission still has a prominent role. Communicating different thresholds allows to reveal not only very recent links, but also older TLs, which allows to evaluate the transmission burden, the impact of transmission control programmes, as well as, reveal transmission hotspots and unanticipated risk factors or community transmission, beyond the limits of contact tracing.

## Continuous pairwise genetic distance distributions reflect sustained transmission over the last decades

The evaluation of local-born cases in the Valencia region revealed continuous clustering across genetic distances, similar to Malawi. In both settings, differentiation between linked and unlinked cases seems arbitrary, as a clear SNP cut-off to delineate genomic transmission could not provide precise results (*Figure 4A*). This contrasts with the results of Oxfordshire, where clustering does not change in the range of 12–150 SNPs (*Figure 4B*). In this sense, the SNP threshold choice used to differentiate transmission from unrelated cases remains challenging even in low-burden settings and provides only tentative information (*Meehan et al., 2019*). An in-depth evaluation of clustering in each setting is needed to understand its particular transmission dynamics. Furthermore, the Valencia region and Malawi also display continuous and sustained TLs over time (*Figure 4C*). Those events outside the genomic transmission clusters likely reflect older contagion chains that still contribute to TB incidence today, as a consequence, clustering is continuous in settings exhibiting this transmission dynamics. The lack of effective past efforts to halt transmission may represent a plausible explanation. Epidemiological data demonstrates that Spain will likely attain a country profile similar to the UK and other low-burden, high-immigration countries. The higher transmission and the older age of transmission chains likely reflects a situation in which Spain suffered from higher disease incidence for most of the 20th century, reflecting its lower socioeconomic status than neighboring countries. The current control strategies in place in the Valencia region meet the WHO's target to reduce TB, including active case findings of close contacts since the 1990s. Improved TB control has led to a continuous drop in case numbers and to an incidence from 22 to 6.4 in the last 20 years. By contrast, Oxfordshire displays a bimodal distribution of clustering across pairwise distances, and also lacked transmission events other than those involving 12 SNP gClusters (*Figure 4*). These results agree with the robust reduction in both disease incidence and transmission that occurred until the beginning of the 1990s in the UK; after that, increased HIV infections, immigration and the emergence of TB drug resistance fueled the expansion of TB (*Glaziou et al., 2018*). In accordance with this data, we dated ongoing transmission in Oxfordshire back to 1993. Our results imply uninterrupted transmission of TB in Valencia region and Malawi and not in Oxfordshire and offer an explanation for the differences in SNP distributions across settings (*Figure 4A and B*).

The main limitations of our analysis are (I) methodological, since only cases with positive cultures can be sequenced. However, we showed that cases included are an accurate representation of the epidemiological characteristics of the populations under study. On the other hand (II), in our analysis of historical transmission, we use tree nodes as 'TLs'' to summarize the number and time of historical transmission. Since both are impossible to quantify, we assume that these TLs represent an estimation of the moment when the strain started to circulate and, at least, one person-to-person transmission event. We also assume that the strains were circulating in the region and not elsewhere and later imported. Finally, we have not attempted to quantify the number of person-to-person transmission; which can only be done using approaches like TransPhylo (*Didelot et al., 2017*; *Xu et al., 2019*). However, this approach is suitable only for recent transmission, i.e. cases within transmission clusters. Thus, TransPhylo is not applicable in our study.

Finally (III), differences in the absolute number of cases in each dataset are irrelevant for comparison, since they all represent population-based studies with the same time-window sampling, thus the majority culture positive cases were included in the analysis. In this sense, the distribution of cases in clusters likely reflects the whole transmission dynamics of the settings.

Our results underscore a primary role for continuous transmission in fueling TB incidence in the Valencia region. Likewise, our results strongly suggest that in this particular setting, community transmission is occurring more frequently rather than in household contacts. All these features are similar to some high-burden settings (*Bjorn-Mortensen et al., 2016*; *Guerra-Assunção et al., 2015*; *López et al., 2020*; *Yang et al., 2018*).

The opposite scenario occurs in other low-burden countries (*Jajou et al., 2018*; *Walker et al., 2014*) where transmission is limited and immigration from high-burden countries, also involving reactivation of the disease, represents the primary driver of incidence (*Meumann et al., 2021*). In addition, reported meta-analysis from historical epidemiological studies suggests that, contrary to current assumptions, MTB infection may not be lifelong, and most people are able to clear it (*Behr et al., 2019*). This further suggests that the prevalence of LTBI is much lower than assumed, and most of the

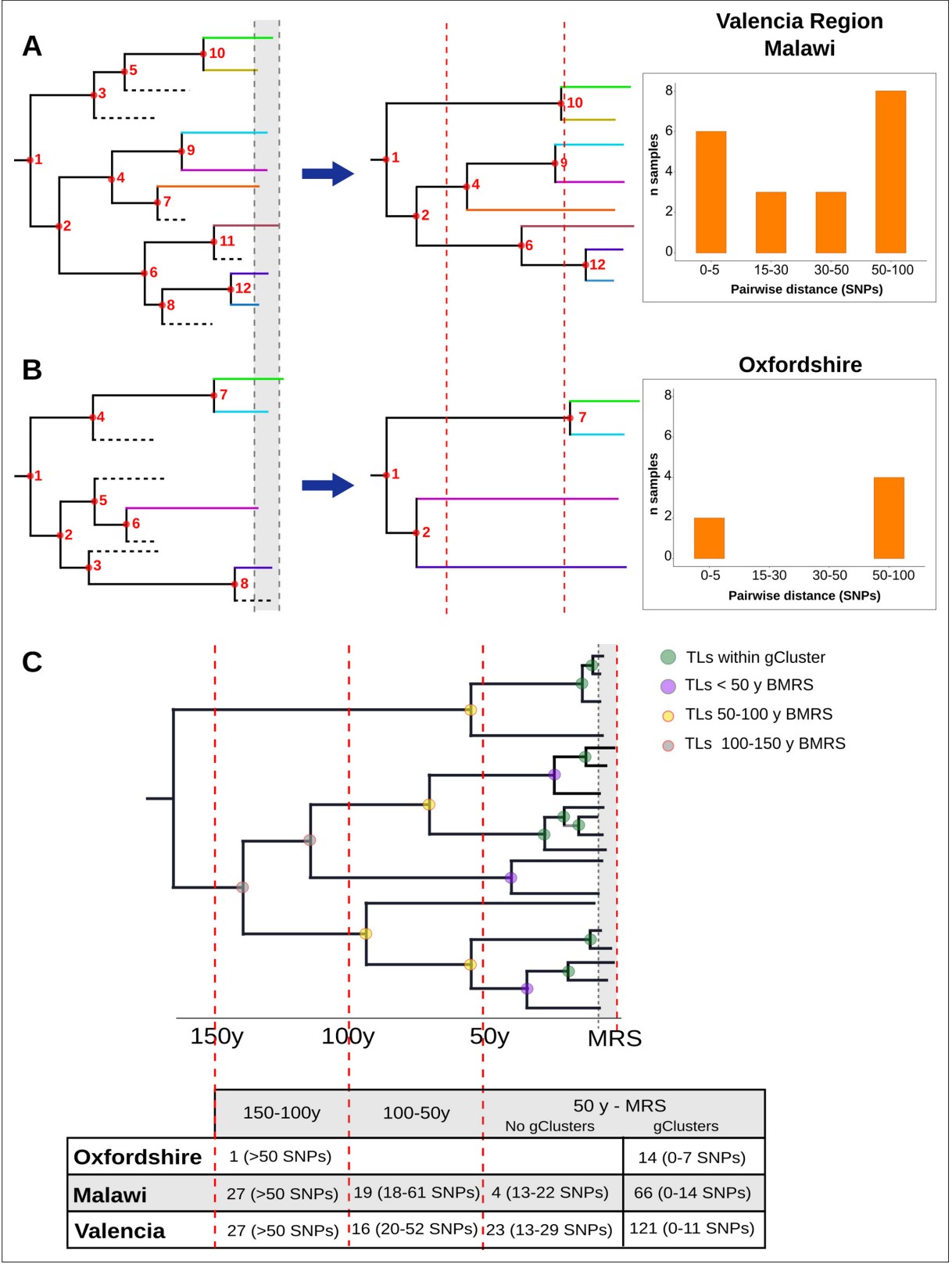

**Figure 4.** Hypothetical time trees indicating transmission links (TLs). (**A**) (Left) The complete phylogeny, including all bacterial isolates and displaying multiple transmission events over time (located at nodes for simplification). This scenario allows the reconstruction of a tree (middle) with several tips and multiple TLs (as the summary of all the events). A continuous distribution of clustered cases by different pairwise distances is retrieved (right) as observed in the Valencia region and Malawi. (**B**) A complete phylogeny (left) in which transmission is either too old or recent and few (or no) transmission

*Figure 4 continued on next page*

*Figure 4 continued*

events occurred in the middle time, led to the reconstruction of a tree (middle) in which few samples reach the present and fewer nodes are observed all over the tree. This scenario provides a bimodal distribution of clustered cases by pairwise distance (right) as observed for Oxfordshire. (**C**) Time tree highlighting TLs over time before the most recent sample (BMRS). The table (bottom) shows the number of links counted in each time period and the median distance range among the samples within the links for the three settings analyzed. For the period between the most recent sample (MRS) and 50 y BMRS, links within (gClusters) and outside gClusters (No gClusters) are indicated. Vertical red lines indicate periods of time, horizontal dashed lines indicate missing samples, shaded areas indicate sampling period, and circles indicate transmission events with colors specified in the legend.

TB cases we see today are coming either from recent contagion or imported depending on the TB setting.

We demonstrate that low burden does not translate to low transmission, highlighting how low-burden TB locations can entail very distinct scenarios that require specifically tailored management in order to eliminate TB, and that general guidelines should not be applied to all the areas based solely on incidence rate (*Lönnroth et al., 2015*). In areas where incidence is mainly contributed by transmission, actions beyond passive case finding strategies are likely to be more successful. Different forms of active case finding to cut community transmission have been implemented in low income countries that can be transferred to high and middle income ones (*Ho et al., 2016*). Those strategies can be designed not only based on the presence of social and host risk factors (*Dowdy et al., 2012*) now there is the opportunity to move toward genomics-informed infection control programmes, e.g., by identifying transmission hotspots or highlighting unanticipated risk factors. Active case finding also has the potential to tackle subclinical TB transmission, which we estimated using high resolution transmission mapping in our setting (*Xu et al., 2019*), and its impact on global and local TB control is still unknown (*Kendall et al., 2021*).

Our key message is that understanding heterogeneities in TB transmission dynamics is essential to define tailor-made interventions to halt transmission with a population-level impact, which is key to reducing the incidence of TB worldwide.

## Additional information

### Group author details

**Valencia Region Tuberculosis Working Group**
**Manuel Belda-Álvarez**: Microbiology Service, Hospital General Universitario de Castellón, Castellón, Spain; **Aurora Blasco**: Microbiology Service, Hospital General Universitario de Castellón, Castellón, Spain; **Avelina Chinchilla-Rodríguez**: Microbiology Service, Hospital General Universitario de Alicante, Alicante, Spain; **Ma Angeles Clari**: Microbiology Service, Hospital Clínico Universitario, Valencia, Spain; **Olalla Martínez-Macías**: Microbiology and Parasitology Service, Hospital Universitario de La Ribera, Alzira, Spain; **Rafael Medina-González**: Microbiology Service, Hospital General Universitario de Alicante, Alicante, Spain; **Fernando Mora-Remón**: Microbiology Service, Hospital General Universitario de Castellón, Castellón, Spain

### Competing interests

Caroline Colijn: Reviewing editor, *eLife*. Valencia Region Tuberculosis Working Group: Iñaki Comas: IC received consultancy fees from Foundation for innovative new diagnostics. The author has no other competing interests to declare. The other authors declare that no competing interests exist.

### Funding

| Funder | Grant reference number | Author |
| --- | --- | --- |
| European Research Council | 638553-TB-ACCELERATE | Iñaki Comas |
| European Research Council | 101001038-TB-RECONNECT | Iñaki Comas |
| Ministerio de Ciencia e Innovación | SAF2016-77346-R | Iñaki Comas |

| Funder | Grant reference number | Author |
| --- | --- | --- |
| Ministerio de Ciencia e Innovación | PID2019-104477RB-I00 | Iñaki Comas |
| European Commission – NextGenerationEU (Regulation EU 2020/2094), through CSIC's Global Health Platform (PTI Salud Global) | SGL2021-03-008 | Iñaki Comas |

The funders had no role in study design, data collection and interpretation, or the decision to submit the work for publication.

## Author contributions

Irving Cancino-Muñoz, Data curation, Formal analysis, Investigation, Visualization, Writing – review and editing; Mariana G López, Conceptualization, Data curation, Formal analysis, Validation, Investigation, Visualization, Methodology, Writing – original draft, Project administration, Writing – review and editing; Manuela Torres-Puente, Validation, Investigation, Methodology, Writing – review and editing; Luis M Villamayor, Resources, Data curation, Methodology, Writing – review and editing; Rafael Borrás, María Borrás-Máñez, Montserrat Bosque, Juan J Camarena, Javier Colomina, Isabel Escribano, Oscar Esparcia-Rodríguez, Ana Gil-Brusola, Concepción Gimeno, Adelina Gimeno-Gascón, Bárbara Gomila-Sard, Damiana Gónzales-Granda, Nieves Gonzalo-Jiménez, María Remedios Guna-Serrano, José Luis López-Hontangas, Coral Martín-González, Rosario Moreno-Muñoz, David Navarro, María Navarro, Nieves Orta, Josep Prat, Juan Carlos Rodríguez, Ma Montserrat Ruiz-García, Resources, Writing – review and editing; Caroline Colijn, Francisco García-García, Formal analysis, Validation, Writing – review and editing; Ester Colomer-Roig, Resources, Methodology, Writing – review and editing; Elvira Pérez, Hermelinda Vanaclocha, Resources, Data curation, Writing – review and editing; Valencia Region Tuberculosis Working Group, Methodology, Resources, Writing – review and editing; Iñaki Comas, Conceptualization, Supervision, Funding acquisition, Investigation, Writing – original draft, Project administration, Writing – review and editing

## Author ORCIDs

Mariana G López (ID) http://orcid.org/0000-0002-2216-9232
Manuela Torres-Puente (ID) http://orcid.org/0000-0002-8352-180X
Rafael Borrás (ID) http://orcid.org/0000-0001-5743-9768
Caroline Colijn (ID) http://orcid.org/0000-0001-6097-6708
Adelina Gimeno-Gascón (ID) http://orcid.org/0000-0003-3728-3182
José Luis López-Hontangas (ID) http://orcid.org/0000-0003-1426-3672
Rosario Moreno-Muñoz (ID) http://orcid.org/0000-0002-0185-5612
Hermelinda Vanaclocha (ID) http://orcid.org/0000-0002-5655-5924
Iñaki Comas (ID) http://orcid.org/0000-0001-5504-9408

## Ethics

Approval for the study was given by the Ethics Committee for Clinical Research from the Valencia Regional Public Health Agency (Comié Ético de Investigación Clínica de la Dirección General de Salud Pública y Centro Superior de Investigación en Salud Pública). Informed consent was waived on the basis that TB detection forms part of the regional compulsory surveillance program of communicable diseases. All personal information was anonymized, and no data allowing patient identification was retained.

## Decision letter and Author response

Decision letter https://doi.org/10.7554/eLife.76605.sa1
Author response https://doi.org/10.7554/eLife.76605.sa2

# Additional files

## Supplementary files

• Supplementary file 1. Samples included in this study and all clinical. Laboratory and epidemiological data associated. WGS data includes all the information obtained by WGS; run

accession number are included in the ENA projects PRJEB29604 and PRJEB38719; depth coverage. MTBC lineage. genomic and epidemiological clusters ID. Patient information including: age, gender, country of birth, year of arrival to Spain, and province of residence in the Valencia region. NA indicates not available epidemiological information.

• Supplementary file 2. Beast input file (.xml) indicating ascertainment bias correction implemented.

• Supplementary file 3. Demographic. clinical and epidemiological data for all culture positive cases and sequenced samples reported between 2014 and 2016 in Valencia region.

• Supplementary file 4. Statistical analysis of clustered vs. unique TB cases. Age, sex, place of birth, sputum smear, and disease type were considered in the analysis for all TB cases and for Spanish-born only.

• Supplementary file 5. Epidemiological cluster evaluation. Clusters identified by epidemiological links were compared against genomic clustering. Similarities and discrepancies between both methods are described for all samples. Cluster size and pairwise distance among the samples included in the cluster are mentioned. Median time and 95% HPD (highest posterior density) of the most common ancestor (tMRCA) for the cluster obtained by both approaches are provided. For mean and median values. Only the clusters with tMRCA were considered.

• Supplementary file 6. Performance of the WGS method for transmission. Values were extracted from 724 clinical TB isolates and assumed epidemiological links from contact tracing as the gold standard. NPV, negative predictive value; PPV, positive predictive value.

• Supplementary file 7. Patients mean age for those cases within genomic clusters which ancestors' age are included in different time windows in Valencia region.

• MDAR checklist

### Data availability

Sequencing data have been deposited in ENA under accession codes PRJEB29604, and PRJEB38719. All data generated or analysed during this study are included in the manuscript and supporting file. Supplemental Tables have been provided with all the analyses performed. All script used and reference sequences are available in http://tgu.ibv.csic.es/?page_id=1794 and https://gitlab.com/tbgenomicsunit/ThePipeline, (copy archived at swh:1:rev:a725827cb664e6d99582 3f3f30fcd1d7e16f63d2).

The following datasets were generated:

| Author(s) | Year | Dataset title | Dataset URL | Database and Identifier |
|---|---|---|---|---|
| Lopez MG | 2019 | Mycobacterium tuberculosis samples to infer transmission clusters | https://www.ebi.ac.uk/ena/browser/view/PRJEB29604 | ENA, PRJEB29604 |
| Lopez MG | 2019 | Population-based study of Mycobacterium tuberculosis samples | https://www.ebi.ac.uk/ena/browser/view/PRJEB38719 | ENA, PRJEB38719 |

The following previously published datasets were used:

| Author(s) | Year | Dataset title | Dataset URL | Database and Identifier |
|---|---|---|---|---|
| Walker et al | 2014 | Assessment of Mycobacterium tuberculosis transmission in Oxfordshire, UK, 2007-12, with whole pathogen genome sequences: an observational study ena | https://www.ncbi.nlm.nih.gov/bioproject/PRJEB5162/ | NCBI BioProject, PRJEB5162 |
| Guerra-Assunção et al | 2015 | Large-scale whole genome sequencing of M. tuberculosis provides insights into transmission in a high prevalence area | https://www.ncbi.nlm.nih.gov/bioproject/?term=PRJEB2358 | NCBI BioProject, PRJEB2358 |

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

## Appendix 1

### Supplemental methods

#### Sample selection and study design

Between 2014 and 2016, the Valencian Regional Public Health Agency (*DGSP: Dirección General de Salud Pública*) reported 1388 TB cases (https://www.sp.san.gva.es/), of which 1019 (78.9%) were culture-positive, as tested by liquid Mycobacteria Growth Indicator Tube (MGIT) or solid-media Löwenstein-Jensen in 18 regional hospitals. 785 single-patient samples were available for further analysis, representing 77% (785/1019) of all culture-positive cases and 60.7% of all active TB cases reported in the region (*Supplementary file 1*, *Appendix 1—figure 1*).

Demographic, clinical, and microbiological records were obtained from the routine TB surveillance system. All diagnosed TB-positive patients completed a standardized questionnaire provided by the DGSP that collected information that included current resident address, country of birth, close contacts, and health status. Census data were obtained from (http://pegv.gva.es/en/noticias/-/asset_publisher/CWK0IEKbs79H/content/padron-municipal-de-habitantes-2016).

#### Genomic cluster definition and WGS phylogenetic inference

A multisequence alignment (MSA) file with the variant positions from all samples was constructed, discarding well-known drug resistance positions. Then, the pairwise SNP distance between all strains was computed with the R *ape* package. Genomic transmission clusters were constructed if the genetic distance between at least two patients' isolates fell below a defined threshold. A customized script was used, and different SNP thresholds considered. To confirm cluster monophyly, a maximum likelihood tree was constructed with the MSA file (50,184 SNPs) using RAxML v8 (*Stamatakis, 2014*), the GTR GAMMA model, 1,000 bootstrap replicates, and including *M. canetti* as an outgroup.

#### Dating recent common ancestors (MRCA) of local transmission clusters

Timed phylogenies for datasets from Malawi (2008–2010), Oxfordshire, and the Valencia Region were inferred with Beast v2.5.1 (*Bouckaert et al., 2014*). First, an MSA file for each dataset was constructed without drug resistance-related SNPs. Since the Valencian dataset was too large, four subsets were defined by considering major clades from the whole phylogeny. Sequences from ancient TB DNA obtained from archeological bones with radiocarbon dating information (*Bos et al., 2014*) and three samples from a recent Spanish outbreak with known time origin (*Pérez-Lago et al., 2019*) were included in all three datasets. Ascertainment bias was corrected by adjusting the clock rate to each alignment length as highlighted in *Supplementary file 2*. A correlation analysis between height median estimated by our correction method and by other commonly used for ascertainment bias correction (https://groups.google.com/g/beast-users/c/QfBHMOqImFE), was performed. As shown in *Appendix 1—figure 4*, correlation between both correction methods is high and significant for each of the dataset analyzed, suggesting that dates on the branch are correctly estimated and comparable between the independent analysis.

The following priors were used for dating (*López et al., 2020*): GTR +GAMMA substitution model, a strict molecular clock model, and a coalescent constant size demographic model. A log-normal prior was used for the clock model with a mutation rate of $4.6 \times 10^{-8}$ (0.20 SNPs per genome per year), reflecting the previously estimated mutation rate of MTBC (*Bos et al., 2014*). Radiocarbon dates were used as prior for ancient samples' tip date, and the Spanish outbreak was included as a calibration node; both cases used a log-normal prior distribution. Three independent runs of Markov Chain Monte-Carlo (MCMC) length chains of 10 million were performed. The runs were combined via logcombiner v2.4.8. Tracer v1.6 was used to determine adequate mixing, convergence of chains and sufficient sampling for every parameter (effective sample sizes, ESS >200), after a 10% burn-in. Trees were annotated with the TreeAnnotator tool. The trees were visualized with FigTree v1.4.4, and the estimated time of the MRCA of every local-born genomic cluster was identified.

### Statistical analysis

Epidemiological information was available for 92% of sequenced samples (724/785). Those cases were used to evaluate associations between risk factors and genomic transmission (clustered cases vs. unique cases) using Fisher's exact test in R. The performance of WGS for detecting transmission was compared against classic contact tracing – to this end, sensitivity, specificity, accuracy, positive predictive values (PPV), and negative predictive values (NPV) were calculated (*Parikh et al., 2008*).

Following the definitions of *Diel et al., 2019*, sensitivity was calculated as the probability of clustering epidemiologically linked cases (true positive value) by WGS. Specificity was calculated as the probability of classifying as unique those epidemiologically unlinked cases (true negative value) by WGS. PPV was calculated as the percentage of patients within a WGS cluster who had an epidemiologically confirmed link. NPV was calculated as the percentage of unique WGS cases without an epidemiologically confirmed link.

A ROC curve was calculated from a subset of the pairwise distance file, including pairwise distances from 0 to 2000 SNPs. We named as a 'cluster' those pairs defined as epidemiologically linked cases by contact tracing and 'unique' those pairs without a link. Data were analyzed and plotted using the pROC R package (*Robin et al., 2011*).

Logistic regression was calculated with R, including all the risk factors and considering clustering as response. Cluster condition was recodified as '1' and unique as '0'. All missing values were eliminated. Model assumptions, including linearity; influential values and multicollinearity were tested and verified in R. Goodness of fit of the logistic regression was tested with Hosmer-Lemeshow test. The best model was defined with the stepwise method, implemented in MASS library; it includes age, gender, place of birth, imprisonment, disease type and homelesness. The Akaike Information Criterion (AIC) indicates that the reduced model is better (AIC = 717.08) than the complete one (AIC = 726.42).

## Supplemental results
### Comparison between epidemiological and genomic clustering
Epidemiological contact tracing was performed for 78% of study cases by DGSP with 97 cases identified as involved in transmission chains (66 epidemiological clusters). Genomic clustering was calculated for all the available sequences, based on a pairwise distance of 12 SNPs for this comparison.

Comparison of both approaches revealed that only 51 (15.4%) of the 331 patients within genomic clusters had an identified epidemiological link. From those, 18 patients displayed complete agreement between both methods, meaning that same clustering was observed for both approaches; and the remaining 33 patients belong to larger genomic clusters, including links not detected by contact tracing.

On the other hand, for the remaining 46 cases with epidemiological links, 27 cases exhibiting disagreement between both methods, either because epidemiological links were different from genomic links (4 cases) or not detected (4 cases). The remaining 19 genomic links could not be evaluated since only one sample from the cluster was sequenced *Supplementary file 5*.

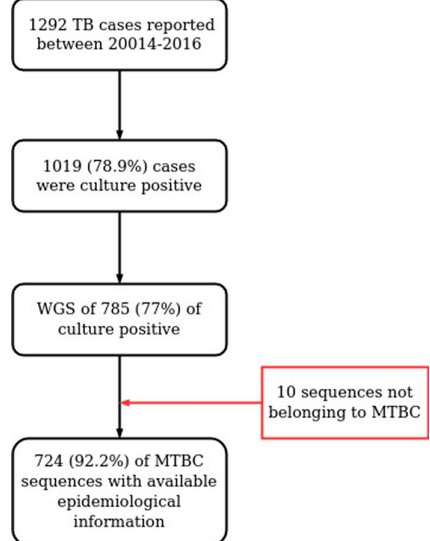

**Appendix 1—figure 1.** Workflow for sample selection. MTBC, *Mycobacterium tuberculosis* complex; WGS, whole genome sequencing.

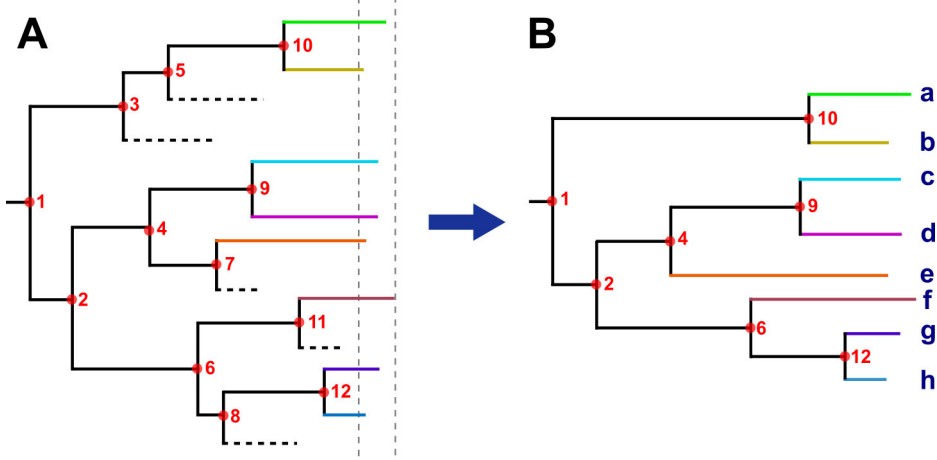

**Appendix 1—figure 2.** Hypothetical time tree. (**A**) Complete phylogeny, including all bacterial isolates. Dashed lines represent missing samples that could not be retrieved during the sampling period (gray dashed lines). Transmission events are indicated as red circles, always in the tree node for simplification; multiple events occurred widely distributed across time phylogeny. (**B**) The tree was reconstructed with the collected samples. Most, but not all, transmission events can be recovered, they were summarized as 'transmission links'. Letters indicate samples collected.

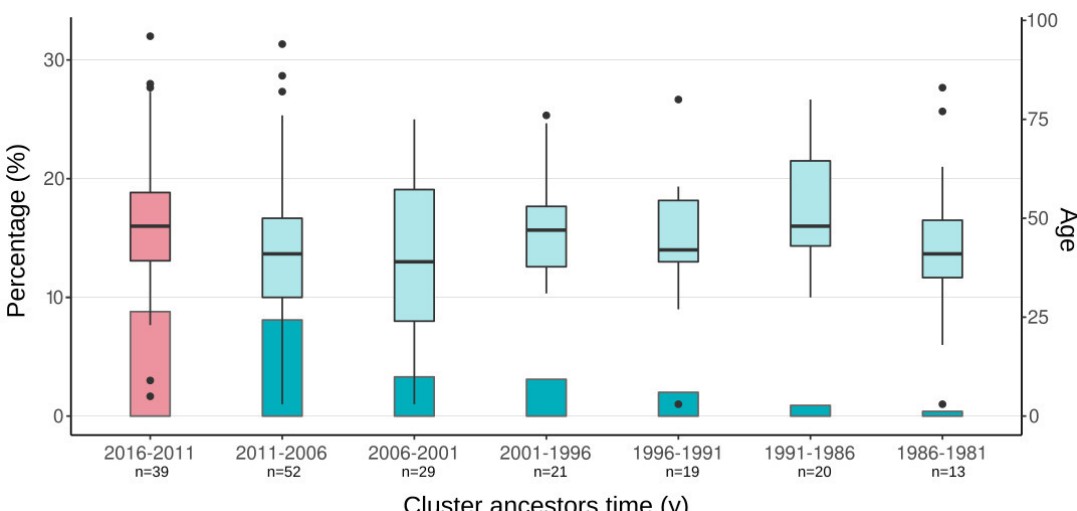

**Appendix 1—figure 3.** Boxplots of ages of cases from spanish-born genomic clusters in the Valencia region vs. the inferred ancestor of the clusters. Bars represent the percentage of cases in gClusters (by 12 SNPs) for each time period. Boxplots represent the age distribution of patients within the clusters. Differences between the age cases for each time period against the most recent clusters (pink) are not significant (Welch two-samples t-test, p-value < 0.1 detailed in *Supplementary file 7*). Sample size of each category is indicated in x-axis label.

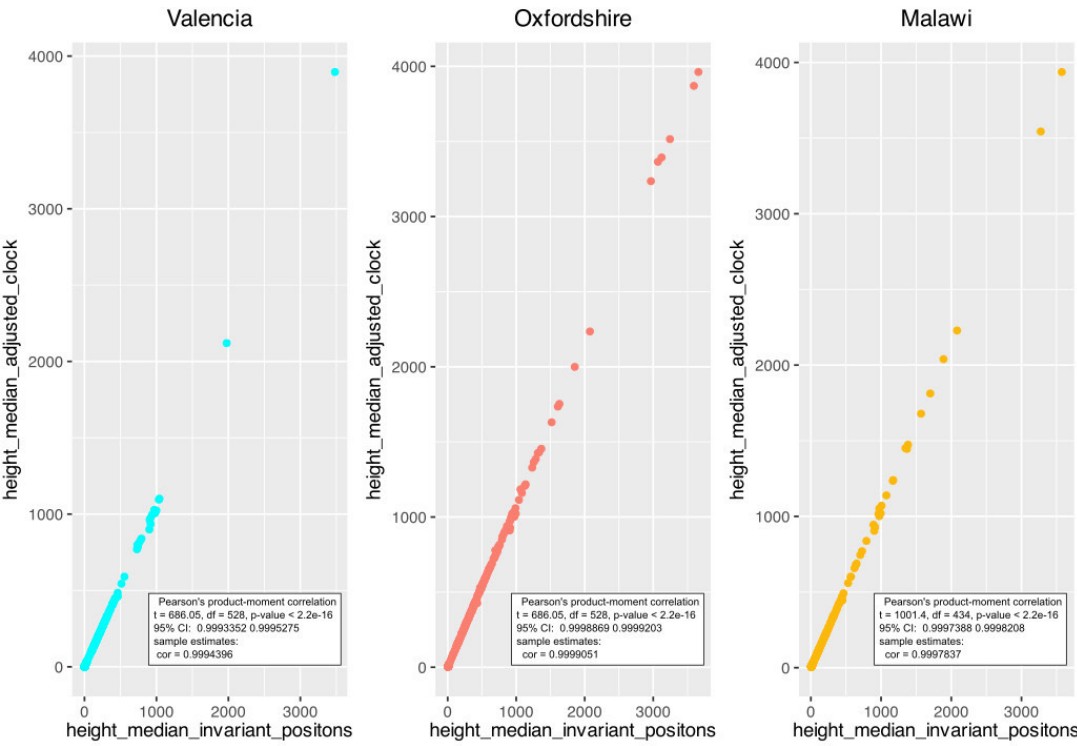

**Appendix 1—figure 4.** Correlation between the height median estimated by two ascertainment bias correction approaches; 'adjusting clock rate' and 'including invariant positions'. Correlation was calculated for each dataset.

