## [Editor Report]

This work presents insightful epidemiologic and phylogenetic analyses of tuberculosis cases across Valencia, Spain, and comparator low-burden (Oxfordshire, UK) and high-burden (Karonga, Malawi) regions. Findings reveal that the "low burden" observed in Valencia is not in fact reflective of low transmission in this setting, with detected lineages likely to have circulated locally over the course of decades and to have been transmitted in the community.

---

## [Decision Letter]

**Decision letter after peer review:**

Thank you for submitting your article "Population-based sequencing of *Mycobacterium tuberculosis* reveals how current population dynamics are shaped by past epidemics" for consideration by *eLife*. Your article has been reviewed by 2 peer reviewers, and the evaluation has been overseen by a Reviewing Editor and a Senior Editor. The following individuals involved in review of your submission have agreed to reveal their identity: Conor J Meehan (Reviewer #1); Timothy M Walker (Reviewer #2).

As is customary in *eLife*, the reviewers have discussed their critiques with one another. What follows below is the Reviewing Editor's edited compilation of the essential and ancillary points provided by reviewers in their critiques and in their interaction post-review. Please submit a revised version that addresses these concerns directly. Although we expect that you will address these comments in your response letter, we also need to see the corresponding revision clearly marked in the text of the manuscript. Some of the reviewers' comments may seem to be simple queries or challenges that do not prompt revisions to the text. Please keep in mind, however, that readers may have the same perspective as the reviewers. Therefore, it is essential that you attempt to amend or expand the text to clarify the narrative accordingly.

Essential revisions:

1) Comments from Reviewer 1 emphasize the need for several refinements to the BEAST analyses, most important among which is the issue of ascertainment biases. If sampling completeness is systematically different over time or in association with particular patient epidemiologic characteristics, then this may raise concerns about interpretation of the findings from this analysis. Specific reference is also made here about distinguishing timing of transmission events vs. time to the most recent common ancestor (coalescent).

2) Reviewer 2 raises the comment that the analysis presented in Table S3 would be stronger if a multivariate analysis had been performed, i.e. controlling for each of the risk factors to determine if they were independently associated with transmission events/clustering.

3) Reviewer 2 also raises points regarding the interpretation of SNP thresholds in settings with differing transmission characteristics, which likewise merit consideration when the authors revise their Discussion.

4) Formatting of the revised manuscript should follow *eLife*'s style guidelines including use of an unstructured abstract. Text in the figures, including in legends and axis labels, is in many places illegibly small and cannot be read on a written page or screen without substantial magnification; this should be fixed.

*Reviewer #1 (Recommendations for the authors):*

Low burden vs low transmission

The framing of the 'low burden doesn't mean low local transmission' needs a better set up and implications explanation. On line 104 this is somewhat set up but it needs to be more clearly stated as an aim of the study. Additionally, what public health officials should do in low burden settings to better use these findings could be explained in the discussion. This would make it easier to read and help people create actionable decisions from these findings.

Line 352 about the future of the Spain TB profile: What backs this up? This is a crux, I think, of what you are trying to get at overall in the paper but it does not come across.

Potential sample bias

You had a few L5 and L6 in your dataset, yet these are more likely to not grow sufficiently within 3 weeks on 7H11, usually requiring 4 weeks. Do you think this could create bias in your sequenced dataset?

Referencing past work

There are a lot of great points made by the authors that perhaps come across as novel but have been shown before and should be referred to. Some examples (but not limited to): Roetzer et al. Plos Medicine 2013 looked extensively at contact tracing vs SNPs; Meehan et al. Ebiomedicine 2018 looked at time vs SNP cut-off; Meumann et al. Lancet Regional Health 2021 looked at time between cases linked genomically and epidemiologically. Some comparisons and general discussion of the findings in the context of these and other papers would be of benefit here.

This dataset was previously published in Xu et al. Plos Medicine 2019 but little reference is made to that paper or its findings. Indeed authors start the discussion stating they present the first population study in Valencia, but that's not exactly true since it was previously published by Xu, although not analysed in this context. Also, an explanation of why a 15 SNP cut-off was used there but a 12 SNP cut-off is used here should be made evident to the reader.

Time tree creation

I don't see in the main or supplemental data anything about ascertainment bias correction for the phylogenies, especially the BEAST analysis. Were SNP alignments or complete WGS data used for the time trees? If SNP, were the invariant counts included to correct the branch lengths? If not, the dates on the branches are most certainly incorrect and need to be redone. If the full genome was used, this needs to be more apparent in the text. Perhaps the BEAST XML file could be supplied via a figshare link or similar.

As stated above, I feel the historical transmission analyses is weak and not well explained. I found it difficult to follow but it seems that only local-born samples were used for this? Although the authors state they are not trying to estimate all transmission in this period, even their approach will result in large inaccuracies in different settings with different sampling fractions and burdens. For example, even a small difference in sampling fraction can have an effect on these transmission event estimations without something like TransPhylo to correct for the sampling fraction (see Séraphin et al. Am J Trop Med Hyg. 2018 for an example). The Xu et al. paper even showed that in Valencia there is a lot of missed index cases from epidemiological data that is found by TransPhylo. These would potentially also all be missed here without the additional Bayesian analyses of TransPhylo over the standard BEAST analyses.

My feeling is that the authors are actually trying to estimate coalescent events, not transmission events. I.e. when did two random Spanish-derived isolates in the current dataset share a common ancestor? If done within sub-lineages or similar, the distributions of these coalescence times and comparisons between settings may better illustrate the point than trying to estimate the amount of transmission within this time period.

*Reviewer #2 (Recommendations for the authors):*

The authors use a Fisher's exact test to assess the relationship between risk factors and clustering. Why was a logistic regression model not used, which would have allowed for a multivariable analysis? This would be preferable as would further assess which risk factors remain significant after correcting for others.

Line 221-2: The argument presented here is that 12 SNPs is less applicable where cases can be clustered between consecutive, ever larger SNP intervals. If the idea behind a threshold is essentially pragmatic, denoting a cut-off where sensitivity and specificity are optimised (as elegantly shown in Figure 2), then how is that impacted by the existence of isolates that are 20, 30, 50 or 100 SNPs apart? Is the argument that the existence of such intermediate distances implies possible epidemiological linkage across them that would inform contact tracing in a helpful way? That would seem unlikely, but it seems to be the implication, namely that it renders the 12 SNP threshold (which is/was epidemiologically defined for contact tracing) less applicable. I would have thought that different patterns are seen in Valencia and Malawi than in Oxfordshire because most of the cases there are endemic rather than imported. Long term transmission dynamics (by which we might understand the survival and expansion of particular clones/strains in a population) are indeed best understood phylogenetically rather than based on thresholds. But it's not clear why the data support 12 SNPs being less informative for contact tracing in Valencia or Malawi. If anything, a lower threshold (e.g. 5 SNPs or less) would make more sense where transmission is more intense, as it would be more specific and avoid investigating links between cases who might in truth be separated by several transmission events (i.e. intermediary cases). In subsequent paragraphs, the authors say just this. I would suggest rethinking how these findings are best interpreted in the text here, or at least successfully rebuffing the points above.

---

## [Author Response]

Essential revisions:1) Comments from Reviewer 1 emphasize the need for several refinements to the BEAST analyses, most important among which is the issue of ascertainment biases. If sampling completeness is systematically different over time or in association with particular patient epidemiologic characteristics, then this may raise concerns about interpretation of the findings from this analysis. Specific reference is also made here about distinguishing timing of transmission events vs. time to the most recent common ancestor (coalescent).

Please see Response to Reviewer 1 P7, P8, P9 and P10.

2) Reviewer 2 raises the comment that the analysis presented in Table S3 would be stronger if a multivariate analysis had been performed, i.e. controlling for each of the risk factors to determine if they were independently associated with transmission events/clustering.

Please see Response to Reviewer 2 P1.

3) Reviewer 2 also raises points regarding the interpretation of SNP thresholds in settings with differing transmission characteristics, which likewise merit consideration when the authors revise their Discussion.

Please see Response to Reviewer 2 P2.

4) Formatting of the revised manuscript should follow eLife's style guidelines including use of an unstructured abstract. Text in the figures, including in legends and axis labels, is in many places illegibly small and cannot be read on a written page or screen without substantial magnification; this should be fixed.

We have now followed the recommendations and fixed the issues.

Reviewer #1 (Recommendations for the authors):Low burden vs low transmissionR1.P1. The framing of the 'low burden doesn't mean low local transmission' needs a better set up and implications explanation. On line 104 this is somewhat set up but it needs to be more clearly stated as an aim of the study.

Thank you for emphasizing this point. We have added paragraphs at the end of the introduction and the Discussion section to highlight the problem and to propose ways to move forward.

Line 118: “Recent transmission significantly contributes to the global TB-burden mostly in the high incidence regions and its control is imperative to achieve the goal of the End TB Strategy (Auld et al., 2018; Guerra-Assunção et al., 2015; Yates et al., 2016). On the contrary, in low-burden countries the assumption is that transmission plays at minor role, supported by the fact that in countries close to the pre-elimination phase (<5/100,000 cases) tuberculosis cases are mainly contributed by re-activations of latent TB infection (LTBI) from recently arrived migrants (Menzies et al., 2018). However, whether the minor role of transmission in pre-elimination phase countries can be extrapolated to other low-burden countries is currently unknown. Understanding the correlation between burden and transmission and country specific dynamics is key if tailor-made control strategies are needed.”

Line 476: “High transmission among Spanish-born individuals is a major contributor to disease burden in Valencia. By contrast, reactivation of infections in imported cases from high-burden settings is the significant driver in other low-burden settings (Jajou et al., 2018; Kamper-Jørgensen et al., 2012; Meumann et al., 2021; Walker et al., 2014). Thus, our results reveal the heterogeneity of the TB epidemic among settings, highlighting the lack of correlation between a region’s TB burden and the level of local transmission.”

R1.P2. Additionally, what public health officials should do in low burden settings to better use these findings could be explained in the discussion. This would make it easier to read and help people create actionable decisions from these findings.

Thanks to the reviewer for highlighting this point. We add a paragraph to discussion with detailed actions that can be taken based on our findings.

Line 604: “We demonstrate that low burden does not translate to low transmission, highlighting how low-burden TB locations can entail very distinct scenarios that require specifically-tailored management in order to eliminate TB, and that general guidelines should not be applied to all areas based solely on incidence rate (Lönnroth et al., 2015). In areas where incidence is mainly contributed by transmission, actions beyond passive case finding strategies are likely to be more successful. Different forms of active case finding to cut community transmission have been implemented in low income countries that can be transferred to high and middle income ones (Ho et al., 2016). Those strategies can be designed not only based on the presence of social and host risk factors (Dowdy et al., 2012) now there is the opportunity to move towards genomics-informed infection control programmes for example by identifying transmission hotspots or highlighting unanticipated risk factors. Active case finding also has the potential to tackle subclinical TB transmission, which we estimated using high resolution transmission mapping in our setting (Xu et al., 2019) and its impact on global and local TB control is still unknown (Kendall et al., 2021).”

R1.P3. Line 352 about the future of the Spain TB profile: What backs this up? This is a crux, I think, of what you are trying to get at overall in the paper but it does not come across.

Thanks to the reviewer as this sentence comes after discussing prospects for TB elimination in Spain with public health officials and colleagues. We have now added a reference to the latest Spanish and specifically Valencia Region report. Contrary to other neighboring countries, Spain had historically higher TB rates which are now slowly matching the EU countries with lowest burden. Rates during the 90s’ were higher than 20/100,000 compared to around 10/100,000 in countries like the United Kingdom or Germany. This is consistent with the lower socioeconomic status of the country as compared to other EU countries for the most part of the XXth century:

**Author response image 1. sa2fig1:** 

Modified from “Informe anual de Tuberculosis de la Comunitat Valenciana 2018” released by Valencia Region Health Department.This is compared with the evolution of TB incidence in United Kingdom

(https://thorax.bmj.com/content/73/8/702)

And the incidence of TB across EU countries in 1995 (as an example: https://www.eurosurveillance.org/content/10.2807/esm.03.01.00110-es):

Potential sample biasR1.P4. You had a few L5 and L6 in your dataset, yet these are more likely to not grow sufficiently within 3 weeks on 7H11, usually requiring 4 weeks. Do you think this could create bias in your sequenced dataset?

Thanks for highlighting this point. We believe there is a negligible impact of *M. africanum* in our results.

1) Large hospitals in the region do MGIT and LJ in parallel (which are grown until week 12) as primary cultures to assure culture positivity. If no MGIT positive sample is available then we sequence from the LJ (sometimes with an additional subculturing step). Thus it is unlikely that culture positive cases of L5 or L6 are missed unless they are co-infecting with another, fast growing, lineage (which cannot be discarded but see below).

2) In addition, the most frequent lineage, higher than 90%, in our dataset is L4 as for Europe in general. The immigration profile of the setting also makes unlikely the importation of *M. africanum* cases. We have studied the country of origin of our cases and the prevalence of L5 and L6 in those countries to estimate the % of cases expected to be L5 or L6 based on this review (de Jong et al., 2010) and revision of evidence for African countries not included in the review (Chaoui et al., 2014; Coscolla et al., 2021). During our study period we expected very low numbers of L5 or L6 cases (n=3) and we have identified 5, suggesting that the number of cases missed were minimal. For a distribution of our study cases based on African countries of origin, see table below.

3) Finally, our analysis of long term transmission is based on local-born individuals and notcontributed by L5 or L6. In conclusion, it is likely that we miss some L5 or L6 isolates, but if there is a bias it is not significant and with a negligible effect over the epidemiological analysis presented in our work.

**Author response table 1. sa2table1:** Total cases expected: 3 cases, cases observed: 5*based on (Chaoui et al., 2014; Coscolla et al., 2021; de Jong et al., 2010; Gehre et al., 2016; Yeboah-Manu et al., 2017).

Origen	#total_patients	%_reported*	cases_expected
MAURITANIA	1	no data	
SENEGAL	7	20	1.4
GUINEA EQ	6	no data	
CAMEROON	1	12	0.12
MALI	4	20	0.8
NIGERIA	2	11	0.2
MOROCCO	47	no Africanum reported	
IVORY	1	22	0.2
ALGERIA	5	no data	

R1.P5. Referencing past workThere are a lot of great points made by the authors that perhaps come across as novel but have been shown before and should be referred to. Some examples (but not limited to): Roetzer et al. Plos Medicine 2013 looked extensively at contact tracing vs SNPs; Meehan et al. Ebiomedicine 2018 looked at time vs SNP cut-off; Meumann et al. Lancet Regional Health 2021 looked at time between cases linked genomically and epidemiologically. Some comparisons and general discussion of the findings in the context of these and other papers would be of benefit here.

Thank you for your suggestions. We included and discussed all the references you mentioned and several more. All were cited in different parts of the manuscript and used for comparison and discussion, following your recommendations.

R1.P6. This dataset was previously published in Xu et al. Plos Medicine 2019 but little reference is made to that paper or its findings. Indeed authors start the discussion stating they present the first population study in Valencia, but that's not exactly true since it was previously published by Xu, although not analysed in this context. Also, an explanation of why a 15 SNP cut-off was used there but a 12 SNP cut-off is used here should be made evident to the reader.

Thank you for giving us the opportunity to clarify this issue.

In the paper of Xu et al., we published only part of the dataset used, only cases that were in clusters. Thus, the Xu et al. paper does not describe relationships between all the strains in the dataset including those unique and therefore the present paper is the first describing the transmission profile of the region using genomic epidemiology.

On the other hand, in this new work we present a population-based study of the Valencia Region. One of the main objectives of the study was to evaluate SNP cut-offs in the light of epidemiological links via an ROC curve. We show that a 11.5 SNP cut-off maximizes the number of links without compromising specificity. Thus, we use a 12 SNP threshold. In Xu et al. 2019 the objectives were different and focused on understanding person-toperson transmission not the burden of transmission in the region. For this, we customized Transphylo, a tool that allows reconstructing transmission trees from phylogenetic trees. We used a 15-SNP threshold for two main reasons. 1. TransPhylo has a particular requirement of cluster size -at least 4 cases-, and genetic distances between cases, to be able to perform estimates, because of that we decided to construct clusters considering a larger distance threshold of 15 SNPs than is usually used. 2. Transphylo is largely agnostic to SNP thresholds, so we decided to include very recent transmission events as well as older transmission events. In any case 42.7% of the cases analyzed in this study were in a distance of 12 SNP or below compared to 43.5% 15 SNP.

However, we agree with the reviewer that the Xu et al. 2019 article could have been introduced better and discuss our results in the light of the previous results derived from part of the present dataset. In accordance with the reviewer point, we add reference to the paper of Xu et al. 2019 several times in introduction and results; it was also mentioned again in Discussion.

R1. P7. Time tree creationI don't see in the main or supplemental data anything about ascertainment bias correction for the phylogenies, especially the BEAST analysis. Were SNP alignments or complete WGS data used for the time trees? If SNP, were the invariant counts included to correct the branch lengths? If not, the dates on the branches are most certainly incorrect and need to be redone. If the full genome was used, this needs to be more apparent in the text. Perhaps the BEAST XML file could be supplied via a figshare link or similar.

Thank you for giving us the opportunity to fully clarify the methodology used.

As the reviewer correctly pointed out, SNPs alignments were used in Beast analysis, but ascertainment bias was indeed corrected by adjusting the clock rate based on the length of the multi-fasta file used for each BEAST analysis. There is an alternative method to introduce ascertainment bias including the invariants sites in the XML file (https://groups.google.com/g/beast-users/c/QfBHMOqImFE). To explore which method performs better we have compared our correction approach with the commonly used method of including invariant position in the xml file. We evaluated the correlation between the median height values of each node estimated with both approaches, for each dataset (Figure 1). The correlation is close to 1 and significant, indicating that our approach is appropriate as an ascertainment bias correction method.

As an independent quality control of our dating approach we included in the original analysis a set of ancient DNA from mummies from Bos et al. (2014) dated by radiocarbon in all the independent analyses (as detailed in Appendix 1). The estimated dating of ancient samples agreed among the analysis of each dataset, suggesting that dates on the nodes are correctly estimated and comparable between independent analyses.

In order to clarify this point in the manuscript, we include a paragraph in Methods under the heading “Genomic clustering and phylogenetic analyses” (Line 236). In addition, we have added the correlation figure in Supplemental methods and detailed the ascertainment bias correction implemented. Following the reviewer suggestion, we include a xml (Supplementary File 2) file highlighting the “adjusting clock rate” correction used.

R1.P8. As stated above, I feel the historical transmission analyses is weak and not well explained. I found it difficult to follow but it seems that only local-born samples were used for this?

Thank you for pointing out the necessity of a detailed explanation to this important point.

Time-trees were reconstructed for all samples, but as is detailed in Figure 3-figures supplement only nodes linking local-born samples were considered for this analysis. Excluding those in which foreign-born samples appeared as the oldest sample (i.e. likely introductions), see Figure 2 where nodes used for the historical transmission analyses are highlighted.

We completely modified the text under the heading “Tracking historical transmission links” (Line 253) in the Methods section; in addition, we changed “events” by “links” throughout the manuscript. We carefully detailed the methodology and the rationale used in this analysis. In addition, we modify the results paragraph under the heading “Historical transmission links between clinical settings highlight distinct epidemic dynamics” (Line 410), including a paragraph to detail the approach used.

Line 412: “In order to evaluate transmission dynamics in a setting over time, we defined historical transmission links (TLs) as the common ancestor of two circulating strains up to 150 years before 2016 (yB 2016). To notice, we did not try to quantify how many transmission events have happened over the last 150 years. Our rationale is that many person-to-person transmission events likely occurred along branches between nodes or nodes and terminals, they are impossible to quantify, but we can summarize all these events as one transmission link, as we are confident that at least one transmission event occurred along the branch. The exact time of the transmission is not possible to estimate either, instead our rationale is that when two circulating strains had a common transmission link in the past, this ancestor represents a lower-bound for when the strains started to circulate. Thus, we compare how many links have occurred during a period of time among different settings, as an approach of long term transmission dynamics analysis. In our approach, we only considered genomic data from local-born patients to avoid the influence of imported genotypes in our analysis.”

R1.P9. Although the authors state they are not trying to estimate all transmission in this period, even their approach will result in large inaccuracies in different settings with different sampling fractions and burdens. For example, even a small difference in sampling fraction can have an effect on these transmission event estimations without something like TransPhylo to correct for the sampling fraction (see Séraphin et al. Am J Trop Med Hyg. 2018 for an example). The Xu et al. paper even showed that in Valencia there is a lot of missed index cases from epidemiological data that is found by TransPhylo. These would potentially also all be missed here without the additional Bayesian analyses of TransPhylo over the standard BEAST analyses.

We respectfully disagree with the reviewer as the analysis from TransPhylo (one of the authors of the original paper is part of this ms) and Beast are different. With TransPhylo we can identify mostly recent transmission links and missing cases in densely sampled clusters, thus it is focused mainly in recent transmission, as was used in Xu et al. 2019. If TransPhylo were to map transmission events onto long branches, it would be doing that using only the information the user enters about the generation interval. If you assume a long generation interval, then the system estimates fewer transmission events, and vice versa.

If we try to run TransPhylo on a tree with several clusters separated by long branches, TransPhylo spends all its time adding and removing transmission events on the long branches where it has no information, and doesn't converge well in the sense of estimation within the clusters. TransPhylo can't simultaneously estimate the timing of transmission, the effective population size in the host (parameter Neg) and the sampling fraction. However, within the densely-sampled clusters it can help. In this sense, adding more recent transmission links was not our intention, since we were interested in long-term transmission dynamics.

With BEAST we use the ancestral nodes as a surrogate of historical transmission links to reconstruct the historical epidemic in each setting. One major limitation is that if any representative derived from those ancestral transmission events is not present in our sampling period (2014-2016), they will be missed in our historical reconstruction. Thus, any transmission lineage that has died will not be reflected. However, this is precisely one of our points about the impact of interventions when we compare Oxfordshire to Valencia or to Malawi.

On the other hand, our approach used population based datasets, in all cases more than 70% of positive culture cases were sequenced; meaning that we recover almost all the transmission that can be measured by WGS. Malawi was the setting with the least percentage of cases sequenced (72% of culture positive) and Oxfordshire the highest (92%). In this sense, if including more cases had any impact in our results, it would affect more to Malawi and Valencia than to Oxfordshire, probably increasing the differences in the historical transmission dynamics observed, making our results even more significant.

We have now included this explanation in the manuscript under “Tracking historical transmission links” heading (Material and Methods, line 253). And also, in the limitations paragraph into discussion (Line 570).

R1.P10. My feeling is that the authors are actually trying to estimate coalescent events, not transmission events. I.e. when did two random Spanish-derived isolates in the current dataset share a common ancestor? If done within sub-lineages or similar, the distributions of these coalescence times and comparisons between settings may better illustrate the point than trying to estimate the amount of transmission within this time period.

The reviewer is right in the sense that we are estimating coalescent events of samples. However, given that *M. tuberculosis* is an obligate human pathogen those coalescent events should be a good surrogate, or at least a lower bound, of historial transmission events.

We are not clear that repeating the analysis within sublineages will have an impact in our estimations. This is because here we are tracking historical transmission links within a sublineage, they never occur between sublineages as we are looking at events that happened less than 150 years ago. In addition, the dynamics of the sublineages across settings are not comparable because the prevalence and historical impact of those sublineages have been different in each setting.

We add a detailed paragraph to clarify this concept and our rationale in Material and Methods, but also in results and discussion.

Reviewer #2 (Recommendations for the authors):R2.P1. The authors use a Fisher's exact test to assess the relationship between risk factors and clustering. Why was a logistic regression model not used, which would have allowed for a multivariable analysis? This would be preferable as would further assess which risk factors remain significant after correcting for others.

Thank you for your suggestion. We now include the results of logistic regression in Supplementary File 4, we have added a paragraph in the Result section under “Epidemiological and genomic clustering” heading, and we also include a detailed explanation of the methodology used in Appendix 1 under “Statistical analysis” section (line 74). In the multivariable analysis “Place of birth” is still a risk factor of clustering after correcting for other confounding factors.

Line 320: “In addition to Spanish origin, pulmonary localization of TB (OR 2.5, CI 1.60-3.98, p<0.001), and younger age also appeared associated with clustering by Fisher’s exact test. After correcting for confounders using a logistic regression, Spanish origin remains significantly associated with clustering (p<0.001); younger age, pulmonary localization of TB, and male sex were also significant (p<0.05, Supplementary file 4). Finally, 90% of TB cases in Valencia Region are susceptible to all antibiotics used in treatment, so resistance has no impact on clustering.”

R2.P2. Line 221-2: The argument presented here is that 12 SNPs is less applicable where cases can be clustered between consecutive, ever larger SNP intervals. If the idea behind a threshold is essentially pragmatic, denoting a cut-off where sensitivity and specificity are optimised (as elegantly shown in Figure 2), then how is that impacted by the existence of isolates that are 20, 30, 50 or 100 SNPs apart? Is the argument that the existence of such intermediate distances implies possible epidemiological linkage across them that would inform contact tracing in a helpful way? That would seem unlikely, but it seems to be the implication, namely that it renders the 12 SNP threshold (which is/was epidemiologically defined for contact tracing) less applicable. I would have thought that different patterns are seen in Valencia and Malawi than in Oxfordshire because most of the cases there are endemic rather than imported. Long term transmission dynamics (by which we might understand the survival and expansion of particular clones/strains in a population) are indeed best understood phylogenetically rather than based on thresholds. But it's not clear why the data support 12 SNPs being less informative for contact tracing in Valencia or Malawi. If anything, a lower threshold (e.g. 5 SNPs or less) would make more sense where transmission is more intense, as it would be more specific and avoid investigating links between cases who might in truth be separated by several transmission events (i.e. intermediary cases). In subsequent paragraphs, the authors say just this. I would suggest rethinking how these findings are best interpreted in the text here, or at least successfully rebuffing the points above.

Thank you for highlighting this point. We totally agree with the reviewer although we may have failed to pass through the message. There are two important roles for thresholds in our opinion and experience working with public health. On one hand, communicating genomic clustering helps in on-going public health investigations particularly within 5 SNP but we have found cases up to 12 SNP that have clarified the origin of an outbreak. In fact, report 5, 12, 15 SNP thresholds to public health together with a degree of confidence, it is the call from public health officials to pursue or not the potential links. But it is also true that 5 SNP threshold renders the more actionable results.

On the other hand, different thresholds allows to reveal not only very recent transmission (05) but also older transmission events (6-12 SNP) what allows to evaluate the transmission burden, the impact of transmission control programmes as well as reveal transmission hotspots and unanticipated risk factors or community transmission beyond CT limits. This is best exemplified by the differences seen in two low burden settings like Oxfordshire and Valencia. We have now made clear the distinction and the value of the thresholds in our setting.

Following the reviewer suggestion we add the following paragraph:

Line 514: “Communicating different thresholds allows to reveal not only very recent links, but also older transmission links, which allows to evaluate the transmission burden, the impact of transmission control programmes, as well as reveal transmission hotspots and unanticipated risk factors or community transmission, beyond the limits of contact tracing.”

References

Chaoui I, Zozio T, Lahlou O, Sabouni R, Abid M, El Aouad R, Akrim M, Amzazi S, Rastogi N, El Mzibri M. 2014. Contribution of spoligotyping and MIRU-VNTRs to characterize prevalent *Mycobacterium tuberculosis* genotypes infecting tuberculosis patients in Morocco. *Infect Genet Evol* 21. doi:10.1016/j.meegid.2013.05.023

Coscolla M, Gagneux S, Menardo F, Loiseau C, Ruiz-Rodriguez P, Borrell S, Otchere ID, Asante-Poku A, Asare P, Sánchez-Busó L, Gehre F, N’Dira Sanoussi C, Antonio M, Affolabi D, Fyfe J, Beckert P, Niemann S, Alabi AS, Grobusch MP, Kobbe R, Parkhill J,

Beisel C, Fenner L, Böttger EC, Meehan CJ, Harris SR, de Jong BC, Yeboah-Manu D, Brites D. 2021. Phylogenomics of Mycobacterium africanum reveals a new lineage and a complex evolutionary history. *Microbial Genomics* 7:000477.

de Jong BC, Antonio M, Gagneux S. 2010. Mycobacterium africanum—Review of an Important Cause of Human Tuberculosis in West Africa. *PLoS Negl Trop Dis* 4. doi:10.1371/journal.pntd.0000744

Gehre F, Kumar S, Kendall L, Ejo M, Secka O, Ofori-Anyinam B, Abatih E, Antonio M, Berkvens D, de Jong BC. 2016. A Mycobacterial Perspective on Tuberculosis in West Africa: Significant Geographical Variation of M. africanum and Other *M. tuberculosis* Complex Lineages. *PLoS Negl Trop Dis* 10:e0004408.

Hall MD, Woolhouse MEJ, Rambaut A. 2016. Using genomics data to reconstruct transmission trees during disease outbreaks. *Rev Sci Tech* 35:287–296.

Lönnroth K, Migliori GB, Abubakar I, D’Ambrosio L, de Vries G, Diel R, Douglas P, Falzon D, Gaudreau M-A, Goletti D, González Ochoa ER, LoBue P, Matteelli A, Njoo H, Solovic I, Story A, Tayeb T, van der Werf MJ, Weil D, Zellweger J-P, Abdel Aziz M, Al Lawati MRM, Aliberti S, Arrazola de Oñate W, Barreira D, Bhatia V, Blasi F, Bloom A, Bruchfeld J, Castelli F, Centis R, Chemtob D, Cirillo DM, Colorado A, Dadu A, Dahle UR, De Paoli L, Dias HM, Duarte R, Fattorini L, Gaga M, Getahun H, Glaziou P, Goguadze L, Del Granado M, Haas W, Järvinen A, Kwon G-Y, Mosca D, Nahid P, Nishikiori N, Noguer I, O’Donnell J, Pace-Asciak A, Pompa MG, Popescu GG, Robalo Cordeiro C, Rønning K, Ruhwald M, Sculier J-P, Simunović A, Smith-Palmer A, Sotgiu G, Sulis G, Torres-DuqueCA, Umeki K, Uplekar M, van Weezenbeek C, Vasankari T, Vitillo RJ, Voniatis C, Wanlin M, Raviglione MC. 2015. Towards tuberculosis elimination: an action framework for low-incidence countries. *Eur Respir J* 45:928–952.

Yeboah-Manu D, de Jong BC, Gehre F. 2017. The Biology and Epidemiology of. Strain Variation in the Mycobacterium tuberculosis Complex: Its Role in Biology, Epidemiology and Control 117–133.